# Loss of tumor suppressor TMEM127 drives RET-mediated transformation through disrupted membrane dynamics

Timothy J Walker[1], Eduardo Reyes-Alvarez[1], Brandy D Hyndman[1], Michael G Sugiyama[2], Larissa CB Oliveira[1], Aisha N Rekab[1], Mathieu JF Crupi[1], Rebecca Cabral-Dias[2], Qianjin Guo[3], Patricia LM Dahia[3], Douglas S Richardson[4], Costin N Antonescu[2], Lois M Mulligan[1]*

[1]Division of Cancer Biology and Genetics, Cancer Research Institute, and Department of Pathology and Molecular Medicine, Queen's University, Kingston, Canada; [2]Department of Chemistry and Biology, Toronto Metropolitan University, Toronto, Canada; [3]Division of Hematology and Medical Oncology, University of Texas Health Science Center, San Antonio, United States; [4]Department of Molecular and Cellular Biology, Harvard Center for Biological Imaging, Scientific Image Analysis Group, Harvard University, Cambridge, United States

*For correspondence:
mulligal@queensu.ca

Competing interest: The authors declare that no competing interests exist.

**Abstract** Internalization from the cell membrane and endosomal trafficking of receptor tyrosine kinases (RTKs) are important regulators of signaling in normal cells that can frequently be disrupted in cancer. The adrenal tumor pheochromocytoma (PCC) can be caused by activating mutations of the rearranged during transfection (RET) receptor tyrosine kinase, or inactivation of TMEM127, a transmembrane tumor suppressor implicated in trafficking of endosomal cargos. However, the role of aberrant receptor trafficking in PCC is not well understood. Here, we show that loss of TMEM127 causes wildtype RET protein accumulation on the cell surface, where increased receptor density facilitates constitutive ligand-independent activity and downstream signaling, driving cell proliferation. Loss of TMEM127 altered normal cell membrane organization and recruitment and stabilization of membrane protein complexes, impaired assembly, and maturation of clathrin-coated pits, and reduced internalization and degradation of cell surface RET. In addition to RTKs, TMEM127 depletion also promoted surface accumulation of several other transmembrane proteins, suggesting it may cause global defects in surface protein activity and function. Together, our data identify TMEM127 as an important determinant of membrane organization including membrane protein diffusability and protein complex assembly and provide a novel paradigm for oncogenesis in PCC where altered membrane dynamics promotes cell surface accumulation and constitutive activity of growth factor receptors to drive aberrant signaling and promote transformation.

## eLife assessment

This **valuable** paper provides **convincing** evidence that loss of the tumor suppressor TMEM127 causes disorganization of plasma membrane lipid domains, alters clathrin assembly, and inhibits endocytosis of a variety of cell surface receptors, leading to increased cell surface levels of signaling proteins including RET and other transmembrane receptor tyrosine kinases. The results are significant for understanding how RET127 loss contributes to pheochromocytoma, although the evidence is indirect owing to the lack of human pheochromocytoma cell lines. The results will be of interest for researchers studying pheochromocytoma and endocytosis mechanisms.

## Introduction

Internalization of activated receptor tyrosine kinases (RTKs) from the cell membrane and trafficking through endosomal compartments is a critical mechanism regulating the magnitude and duration of downstream signals (*Schmid, 2017*). RTKs internalized by clathrin-mediated endocytosis (CME) transition through successive early and late endosomes for eventual degradation in lysosomes (*Sorkin and Fortian, 2015*). In cells where endocytosis or trafficking is impaired, RTKs may accumulate inappropriately at the cell surface or in aberrant subcellular compartments, contributing to cell transformation and oncogenic growth. Despite the importance of CME and RTK cargo transition through compartments, the trafficking proteins required for transitions and their roles in these processes remain poorly understood.

RET (rearranged during transfection) is an RTK required for the development of the kidneys, nerves, and neuroendocrine cell types (*Mulligan, 2018*). Under normal conditions, RET is activated by binding of its glial cell line-derived neurotrophic factor (GDNF) ligands and GPI-linked GDNF family receptor α (GFRα) coreceptors, which together recruit RET complexes to lipid rafts, leading to downstream signaling through multiple pathways (*Mulligan, 2018*; *Pierchala et al., 2006*). Activated RET at the cell membrane is internalized via CME to early endosomes where signaling persists, and subsequently trafficked through the endolysosomal system for downregulation and degradation (*Crupi et al., 2015*; *Richardson et al., 2012*). Activating mutations of RET or overexpression of the wildtype (WT) protein are established drivers in several cancers (*Mulligan, 2018*). Constitutively active RET mutants give rise to the inherited cancer syndrome multiple endocrine neoplasia type 2 (MEN2), which is characterized by the adrenal tumor pheochromocytoma (PCC), while activating mutations or increased RET expression are found in a subset of sporadic PCC (*Le Hir et al., 2000*; *Mulligan, 2018*; *Takaya et al., 1996a*; *Takaya et al., 1996b*).

PCC has also been associated with inactivating mutations of TMEM127, a multipass integral membrane protein broadly expressed at low levels and localized to the plasma membrane, endosomes, and lysosomes in normal cells (*Deng et al., 2018*; *Qin et al., 2014*; *Qin et al., 2010*). TMEM127 functions are poorly understood but it has been suggested to facilitate endosomal transition and cargo trafficking and has been shown to modulate endolysosomal function in renal cell carcinomas (*Deng et al., 2018*; *Qin et al., 2010*). Loss-of-function mutations of *TMEM127* have been identified in both familial and sporadic PCC, leading to loss of TMEM127 protein or mislocalization to the cytosol in tumor cells (*Neumann et al., 2019*; *Qin et al., 2010*). Interestingly, our recent data have demonstrated that RET expression is increased in TMEM127-mutant PCC (*Guo et al., 2023*).

The presentation of RET-mutant or TMEM127-deficient PCC are not clinically distinct. Both RET and TMEM127 have been predicted to confer PCC susceptibility through altered kinase signaling (*Mulligan, 2018*; *Neumann et al., 2019*; *Qin et al., 2010*), however, their relationship has not been explored. The importance of endolysosomal trafficking in regulating the intensity and duration of RET pro-proliferative signals suggested that loss of TMEM127-mediated endosomal transition could alter RET trafficking, promoting accumulation or signaling from aberrant subcellular compartments and contributing to transformation. In this study, we have investigated the effects of TMEM127 depletion on RET regulation and function and more broadly on cellular processes that could contribute to PCC pathogenesis. Here, we show that loss of TMEM127 leads to cell surface accumulation and constitutive activation of RET and that this accumulation is due to alterations in membrane organization and composition that reduce efficiency of CME and decrease internalization of RET as well as other transmembrane proteins, suggesting a global impairment of membrane trafficking. As a result of changes in membrane dynamics, TMEM127 depletion increased RET half-life and led to constitutive RET-mediated signaling and cell proliferation that was blocked by RET inhibition. Our data further demonstrate that TMEM127 loss increases membrane protein diffusability and impairs normal membrane transitions and the stabilization and assembly of membrane protein complexes, allowing inappropriate accumulation of actively signaling RET molecules at the cell membrane, and that mislocalized RET is the pathogenic mechanism in TMEM127-mutant PCC.

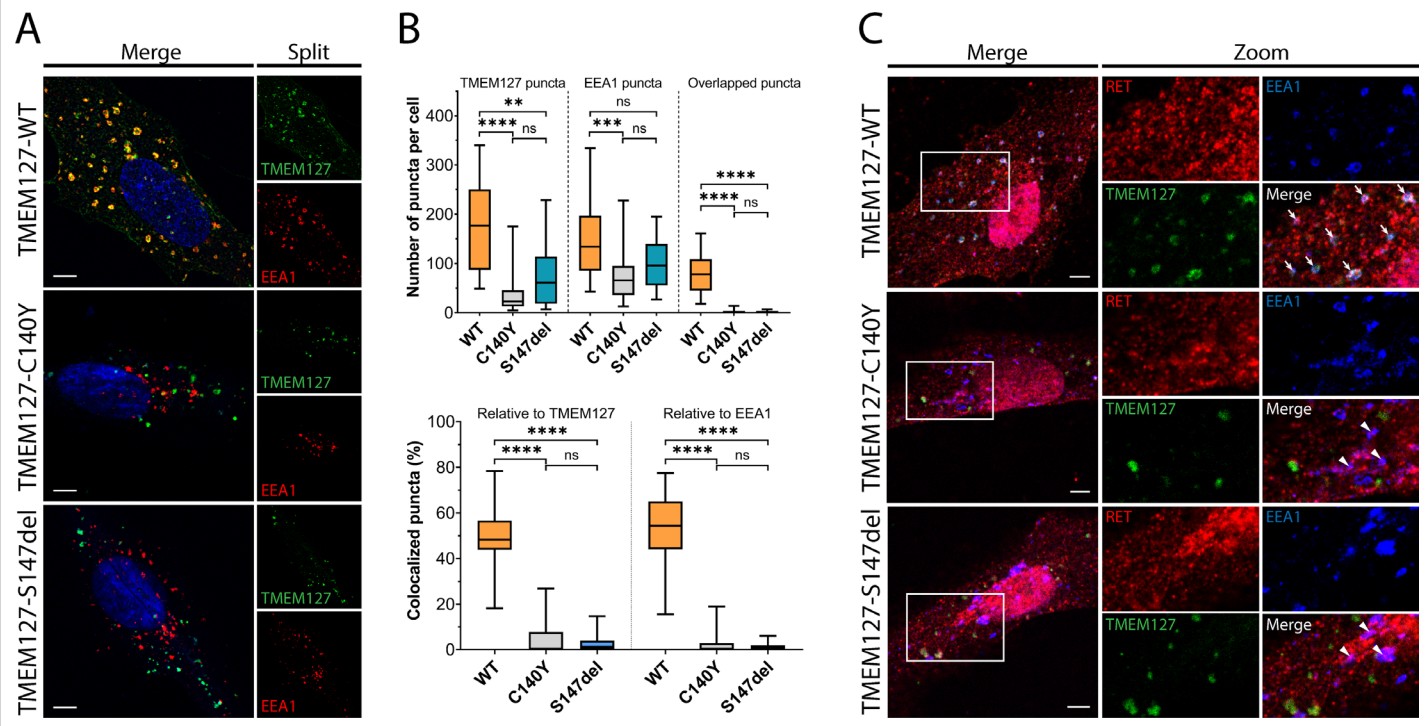

**Figure 1.** Wildtype (WT) and mutant TMEM127 are differentially localized in endosomes. (**A**) Immunofluorescence confocal images of ARPE-19 cells transiently expressing EGFP-TMEM127-WT, -C140Y, or -S147del, and stained for early endosome marker EEA1 (red) and Hoechst nuclear stain (blue). Yellow puncta in the merged images indicate EEA1 colocalization with WT but not mutant TMEM127. (**B**) Quantification of puncta in cells expressing the indicated TMEM127 proteins under conditions in (**A**). Quantification of TMEM127 positive or EEA1 positive and overlapping puncta (upper panel) and percent colocalized puncta relative to TMEM127 or EEA1 (bottom panel). Data from two independent experiments representing 17–31 cells per condition are shown (Kruskal-Wallis test and Dunn's multiple comparisons test; **p<0.01, ***p<0.001, ****p<0.0001). Scale bars = 5 µm. (**C**) Immunofluorescence confocal images of ARPE-19 cells transiently expressing rearranged during transfection (RET), GDNF family receptor α1 (GFRα1), and EGFP-TMEM127-WT, -C140Y, or -S147del, treated with glial cell line-derived neurotrophic factor (GDNF) (100 ng/ml) for 10 min and stained for RET (red) and EEA1 (blue). White puncta (white arrows) indicate RET, TMEM127-WT, and EEA1 colocalization. Colocalization of RET and EEA1 but not TMEM127 in cells expressing TMEM127-C140Y or TMEM127-S147del mutants is indicated with white triangles. Representative images from 14 to 25 cells per condition. Scale bars = 5 µm. Zoom box = 15 µm × 22 µm.

The online version of this article includes the following source data for figure 1:

**Source data 1.** Excel file of TMEM127 and EEA1 colocalization quantification analyses shown in *Figure 1B*.

## Results

### Loss-of-function mutations disrupt TMEM127 localization and reduce early endosome formation

WT-TMEM127 has previously been shown to localize to endosomal compartments, while loss-of-function PCC-TMEM127 mutants reduce protein expression or may be aberrantly localized in the cytosol (*Flores et al., 2020*; *Qin et al., 2010*). Here, we assessed colocalization of transiently expressed TMEM127 with the early endosome marker EEA1 in human retinal pigment epithelial ARPE-19 cells by immunofluorescence microscopy. We showed that WT-TMEM127 localized robustly to EEA1-positive endosomes but that colocalization was significantly reduced for two transmembrane domain TMEM127 mutants (C140Y, S147del), which did not show appreciable localization to EEA1 puncta (*Figure 1A and B*). Interestingly, the total numbers of EEA1 puncta in cells were modestly reduced in the presence of TMEM127-S147del and significantly reduced for TMEM127-C140Y (*Figure 1B*), suggesting a systemic disruption of endosomal formation or stability in response to loss of TMEM127 function in these cells. In ARPE-19 cells transiently expressing RET and TMEM127, GDNF stimulation promoted colocalization of RET with WT but not mutant TMEM127 in EEA1-positive endosomes (*Figure 1C*). RET and EEA1 colocalization was reduced but not eliminated in the presence of TMEM127 mutants, suggesting TMEM127 loss may limit RET localization to early endosome compartments.

## TMEM127 depletion promotes cell surface RET accumulation

Human PCC cell lines and patient-derived organoid cultures have not been successfully generated to date. We therefore generated a CRISPR-Cas9 TMEM127-knockout (KO) and control (Mock-KO) in human neuroblastoma cell line SH-SY5Y, which endogenously expresses RET and TMEM127 (*Figure 2A*), to explore the effects of TMEM127 loss of function on RET localization and functions. We confirmed that these TMEM127-KO cells had similarly reduced EEA1-positive endosomes to our transient models, described above (*Figure 2—figure supplement 1A*). Interestingly, we observed notably increased levels of RET protein in TMEM127-KO compared to control Mock-KO cells (*Figure 2A*), which was consistent with accumulation of a fully mature glycosylated RET form that is found primarily on the cell surface (*Richardson et al., 2006*; *Richardson et al., 2012*; *Takahashi et al., 1991*). This accumulation was not due to changes in *RET* gene transcription, since TMEM127-KO cells had significantly less RET mRNA in qRT-PCR assays (*Figure 2—figure supplement 1B*), suggesting a possible feedback loop modulating RET transcription. Transient re-expression of WT-TMEM127 in these cells reduced RET protein levels (*Figure 2—figure supplement 1C*). Using a surface protein biotinylation assay (*Figure 2A*, *Figure 2—figure supplement 2*, *Reyes-Alvarez et al., 2022*) to specifically detect plasma membrane proteins, we showed that cell surface RET was significantly increased, by approximately fivefold, in TMEM127-KO compared to Mock-KO SH-SY5Y cells. We also saw similar plasma membrane accumulation of endogenous N-cadherin and transferrin receptor-1 proteins in HEK293 cells depleted for TMEM127, and reintroduction of flag-tagged TMEM127 significantly reduced membrane localization (*Figure 2—figure supplement 1D*). Taken together, the accumulation of RET on the cell surface and reduced RET in early endosomes suggested that TMEM127 loss might disrupt RET internalization.

## TMEM127 depletion promotes RET accumulation through impaired internalization

We assessed RET internalization in response to GDNF over time in TMEM127-KO and Mock-KO cells using a biotinylation internalization assay (*Figure 2B*, *Figure 2—figure supplement 2*, *Reyes-Alvarez et al., 2022*). Consistent with our previous studies (*Crupi et al., 2015*; *Richardson et al., 2012*), minimal RET internalization was detected by 5 min of GDNF treatment in Mock-KO cells and increased over time, with significantly greater internalization detected by 30 min (*Figure 2B*). In TMEM127-KO cells, RET internalization was similar at all timepoints and did not increase over time with longer GDNF treatments, with significantly less internalization observed after 30 min compared to Mock-KO controls (*Figure 2B*). Recycling of internalized RET to the cell surface in biotinylation assays was also reduced in TMEM127-KO compared to Mock-KO cells with a smaller proportion of the internalized RET being returned to the cell surface (*Figure 2—figure supplement 2*), suggesting RET surface accumulation was not due to enhanced recycling. Our data show that RET internalization into TMEM127-KO cells is not GDNF responsive and remains at a low constitutive rate despite high levels of RET protein accumulating on the cell surface, suggesting that TMEM127 depletion may impair cellular internalization mechanisms, such as CME.

## RET localization with cell surface clathrin and internalization are reduced in TMEM127-depleted cells

Previous studies have shown that, in response to ligand, RET is recruited to clathrin clusters assembling on the plasma membrane and internalized into endosomal compartments via clathrin-coated pits (CCPs) (*Crupi et al., 2020*; *Crupi et al., 2015*; *Richardson et al., 2006*). To further explore the surface accumulation of RET and impairment of RET internalization in TMEM127-depleted cells, we assessed RET colocalization with cell surface clathrin by total internal reflection fluorescence microscopy (TIRFM), as previously described (*Aguet et al., 2013*; *Crupi et al., 2020*; *Crupi et al., 2015*; *Hyndman et al., 2017*). In Mock-KO cells, GDNF treatment significantly increased colocalization of RET and plasma membrane clathrin structures (*Figure 3A and B*) and caused a significant loss of cell surface RET intensity (*Figure 3A and B*, *Figure 4A*), suggesting activated RET is being recruited to CCPs and then moving out of the TIRF field as it is internalized into the cell. In contrast, when normalized for the increased total RET intensity on the cell surface, RET-clathrin colocalization in untreated TMEM127-KO cells was significantly less than in untreated Mock-KO cells and was not increased by GDNF treatment (*Figure 3A and B*), suggesting impaired recruitment of RET to cell surface

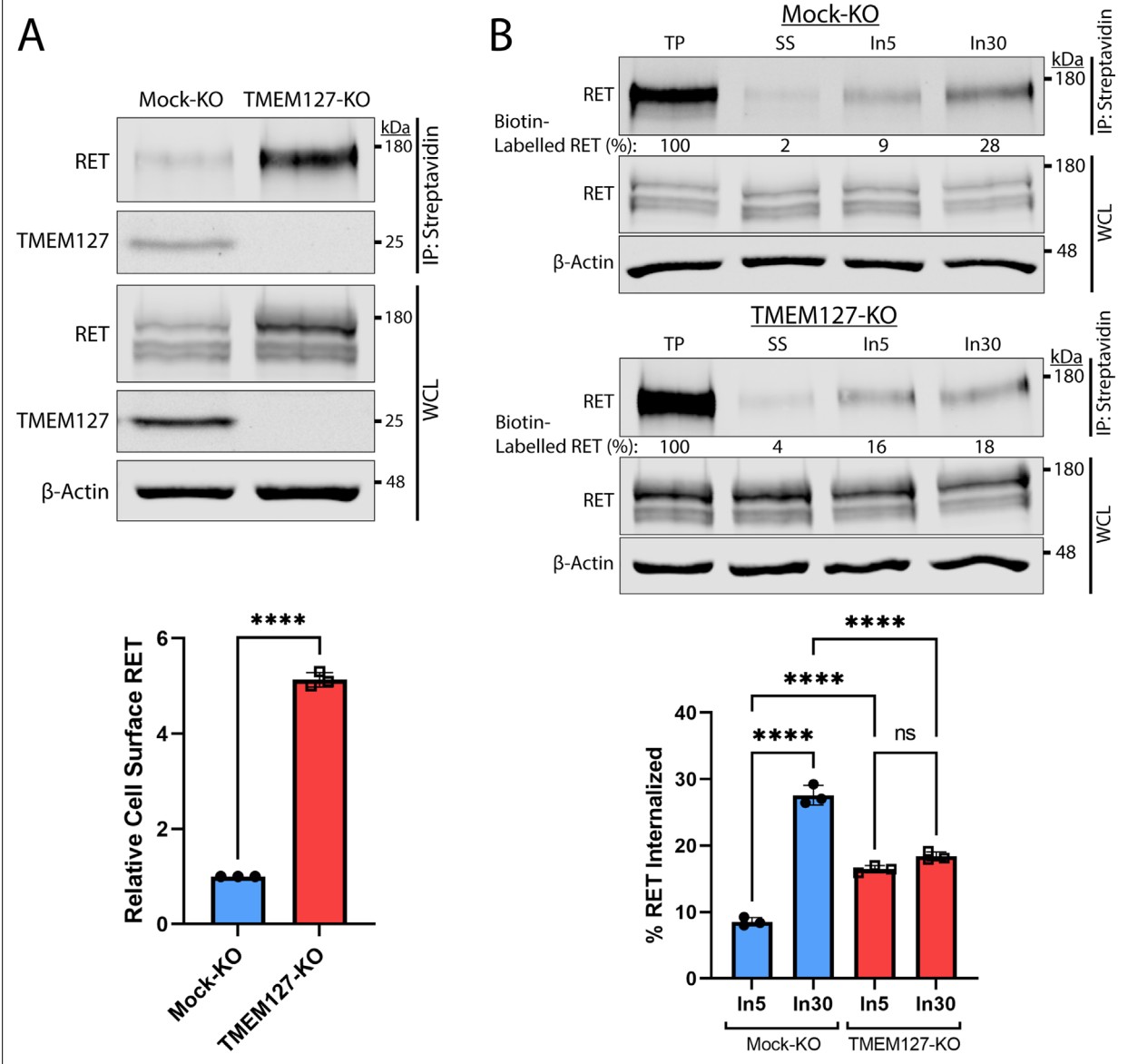

**Figure 2.** Rearranged during transfection (RET) accumulates on the cell surface due to reduced internalization in the absence of TMEM127. (**A**) Immunoblot showing total and cell surface biotinylated RET and TMEM127 proteins in Mock-KO and TMEM127-KO SH-SY5Y cells. Surface proteins were biotinylated and collected by streptavidin immunoprecipitation (IP: streptavidin). Whole cell lysate (WCL) and IP samples were separated by SDS-PAGE and immunoblotted for RET and TMEM127 (upper panel). β-Actin was used as a loading control. Biotinylated cell surface RET levels (IP) were normalized to total RET (WCL) and expressed relative to Mock-KO (bottom panel). Fold increase in relative surface RET is indicated for TMEM-KO cells. Three independent experiments (n=3) are shown as mean ± SD (two-tailed unpaired t-test; ****p<0.0001). (**B**) Immunoblots showing internalization of biotinylated cell surface RET protein in Mock-KO and TMEM127-KO SH-SY5Y cells (top). Surface proteins were biotin labeled (TP, total protein) and either cell surface biotin stripped (SS, surface strip control) or incubated at 37°C with glial cell line-derived neurotrophic factor (GDNF) (100 ng/ml) for 5 (In5) or 30 (In30) min to allow internalization, and remaining cell surface biotin was stripped. Biotinylated proteins were collected, separated, and immunoblotted as in (**A**). β-Actin was used as a loading control. Mean ± SD of internalized biotinylated RET, relative to TP of three independent experiments (n=3) are shown in the bottom panel (one-way ANOVA and Tukey's multiple comparisons test; ****p<0.0001).

The online version of this article includes the following source data and figure supplement(s) for figure 2:

**Source data 1.** Excel file of immunoblot quantification analyses shown in *Figure 2*.

**Source data 2.** Original files for immunoblot analysis shown in *Figure 2*.

**Source data 3.** Labeled file for immunoblot analysis shown in *Figure 2*.

**Figure supplement 1.** Characterization of TMEM127-KO cells.

**Figure supplement 1—source data 1.** Excel file of EEA1 puncta, rearranged during transfection (RET) transcript, and immunoblot quantification

*Figure 2 continued on next page*

*Figure 2 continued*

analyses shown in *Figure 2—figure supplement 1*.

**Figure supplement 1—source data 2.** Original files for immunoblot analyses in *Figure 2—figure supplement 1C and D*.

**Figure supplement 1—source data 3.** Labeled file for immunoblot analyses shown in *Figure 2—figure supplement 1C and D*.

**Figure supplement 2.** Rearranged during transfection (RET) recycling is reduced in the absence of TMEM127.

**Figure supplement 2—source data 1.** Excel file of immunoblot quantification analyses shown in *Figure 2—figure supplement 2*.

**Figure supplement 2—source data 2.** Original files for immunoblot analysis in *Figure 2—figure supplement 2*.

**Figure supplement 2—source data 3.** Labeled file for immunoblot analysis shown in *Figure 2—figure supplement 2*.

clathrin irrespective of ligand stimulation. The overall intensity of cell surface clathrin puncta was not affected by GDNF treatment but was significantly less in TMEM127-KO cells compared to Mock-KO (*Figure 3C*). A decrease of clathrin fluorescence intensity in plasma membrane clathrin structures could suggest a decrease in the number of clathrin subunits incorporated per CCP, and thus reflect a change in clathrin assembly or size. Alternatively, as TIRF intensity decays exponentially with distance from the coverslip, a reduction of clathrin intensity in the TIRF image could also indicate an increase in membrane curvature. To distinguish between these possibilities, we used concomitant widefield epifluorescence microscopy to assess clathrin fluorescence intensity at puncta detected within the TIRF images above (*Figure 3D*), as this form of microscopy is not sensitive to changes in fluorescence intensity by the extent of CCP curvature generation. Clathrin fluorescence intensity in epifluorescence images was robustly reduced in clathrin puncta from TMEM127-KO cells, indicating less clathrin and smaller clusters, not more deeply invaginated structures.

We further assessed numbers and sizes of CCPs in TMEM127 KO cells by imaging EGFP-tagged clathrin with structured illumination microscopy (SIM). Because over 60% of CCPs have been measured to be less than 200 µm in size (*Wu et al., 2001*), the approximate diffraction limit of a light microscope, super-resolution techniques are needed to fully resolve these structures for accurate quantitation. Therefore, we used a technique called dual iterative SIM (diSIM, SIM[2]; *Löschberger et al., 2021*) that combines a lattice illumination pattern with deconvolution and standard SIM processing to resolve lateral structures as small as 60–100 nm in the ~525 nm emission range. Initial visual inspection of diSIM images of TMEM127-KO and Mock-KO cells expressing EGFP-tagged clathrin light chain (EGFP-CLC) suggested TMEM127-KO cells had fewer and smaller CCPs (*Figure 3E*). We further quantified the 2D area of over 6500 CCPs in and near the plasma membranes closest to the coverslip in TMEM127-KO and Mock-KO cells. Here, we found TMEM127-KO cells had a 2D CCP area approximately half of Mock-KO cells (0.008 µm$^2$ vs. 0.016 µm$^2$; *Figure 3F*). An area of 0.008 µm$^2$ corresponds to a diameter of approximately 90 nm, within the resolving power of the diSIM technique (*Löschberger et al., 2021*). Finally, we also quantitated the number of CCPs per cell on a single plane near the lower membrane of TMEM127-KO and Mock-KO cells. The added resolving power of diSIM gave us confidence in reaching a more accurate count (*Figure 3G*). Here, we found a significant decrease in CCPs in TMEM127-KO cells relative to Mock-KO cells (213 vs 289). diSIM provides a clear indication that KO of TMEM127 alters the size and quantity of CCPs in a cell.

Together, our data indicate that TMEM127 depletion led to smaller clathrin clusters or CCPs, suggesting that TMEM127 regulates CCP formation, assembly, and/or turnover. In contrast, we observed a significant 1.3-fold increase in intensity of cell surface RET puncta in TMEM127-KO cells relative to Mock-KO cells (*Figure 4A*), consistent with the cell surface accumulation of RET, but did not observe a decrease in RET puncta in response to GDNF stimulation suggesting impaired internalization. As fewer CCPs are present in TMEM127-KO cells, some of these intense RET puncta are likely not in CCPs but may represent other organization of the receptor at the plasma membrane. Interestingly, we saw a similar increase in surface EGFR puncta intensity in TMEM127-KO cells, with a 1.4-fold increase over Mock-KO cells (*Figure 4B*), demonstrating that protein accumulation at the cell surface is not exclusive to RET in these cells. We extended this observation using surface biotinylation assays and showed that multiple classes of transmembrane proteins including RTKs (RET, EGFR), cell adhesion molecules (N-cadherin, integrin beta-3), and carrier proteins (transferrin receptor-1) accumulate on the cell membrane in TMEM127-KO compared to Mock-KO cells (*Figure 4C and D*), suggesting a global effect caused by impaired surface protein internalization. Together, these data suggest that loss of TMEM127 impairs normal membrane processes including formation of clathrin clusters or

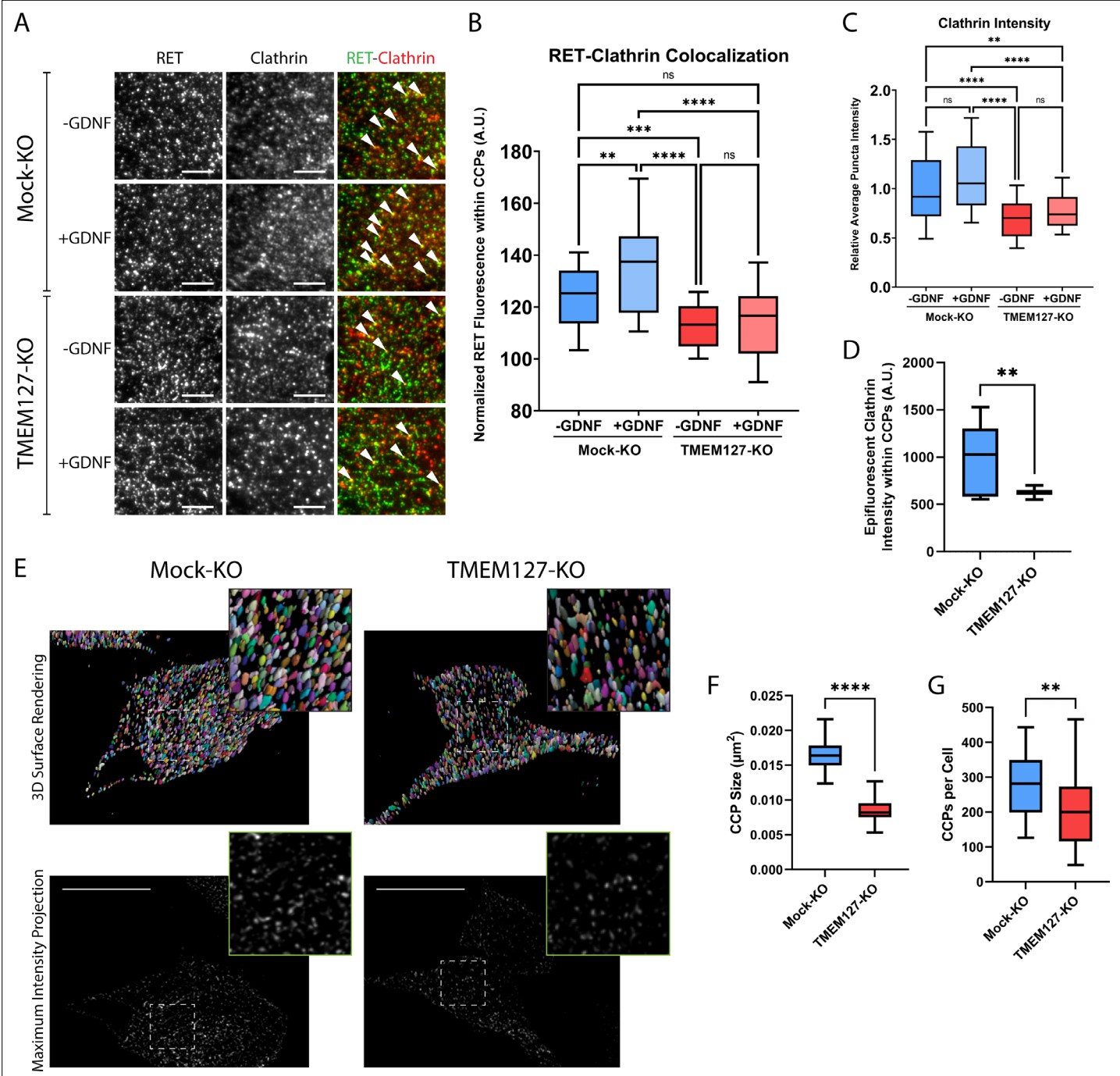

**Figure 3.** Rearranged during transfection (RET) colocalization with cell surface clathrin is reduced and clathrin-coated pits (CCPs) are smaller in TMEM127-depleted cells. (**A**) Representative images of RET and clathrin localization in Mock-KO and TMEM127-KO SH-SY5Y cells treated with (+) or without (-) glial cell line-derived neurotrophic factor (GDNF) (100 ng/ml) for 5 min, fixed and labeled with indicated antibodies, and imaged by total internal reflection fluorescence microscopy (TIRFM). Colocalization of RET and clathrin was detected as yellow puncta (white triangles). Scale bars = 3 µm. (**B–D**) Images represented in (**A**) were subjected to automated detection of CCPs followed by quantification of the mean fluorescence intensity of RET, clathrin, or epifluorescent clathrin therein. (**B**) Mean RET fluorescence within CCPs, normalized to average RET puncta intensity, representing RET and clathrin colocalization. Data from n=30 cells representing a minimum of 5100 CCPs per condition are shown from four independent experiments (Brown-Forsythe and Welch ANOVA tests and Dunnett's T3 multiple comparisons test; **p<0.01, ***p<0.001, ****p<0.0001). (**C**) Average clathrin puncta intensity is shown relative to the Mock-KO (-GDNF) condition. Data from n=50 cells representing a minimum of 8700 CCPs per condition are shown from four independent experiments (Kruskal-Wallis test and Dunn's multiple comparisons test; **p<0.01, ****p<0.0001). (**D**) Mean epifluorescent clathrin intensity within CCPs detected by TIRF, representing intensity extending into the cell, indicating CCP depth. Data from n=34–52 cells per condition are

*Figure 3 continued on next page*

Figure 3 continued

shown from four independent experiments, representing a minimum of 8400 CCPs per condition (two-tailed Mann-Whitney test, **p<0.01). (E–G) Structured illumination microscopy (SIM) imaging of EGFP-tagged clathrin light chain (EGFP-CLC) stably expressed in fixed Mock-KO and TMEM127-KO SH-SY5Y cells. (E) Representative images of 3D renders (upper panel) of entire cells of maximum intensity projections (lower panel) of CCPs in and near the membrane closest to the coverslip. Scale bars = 10 μm. (F) CCP area was measured and plotted (μm²) and (G) number of CCPs per cell was also calculated for n=26–55 cells representing a minimum of 6800 CCPs per condition from three independent experiments (two-tailed Mann-Whitney test; **p<0.01, ****p<0.0001).

The online version of this article includes the following source data for figure 3:

**Source data 1.** Excel file of total internal reflection fluorescence (TIRF) data and clathrin-coated pit (CCP) quantification analyses shown in *Figure 3*.

CCPs and recruitment of cargo, resulting in altered cell membrane composition and accumulation of multiple cell surface proteins.

## TMEM127 depletion impairs assembly and maturation of CCPs

To explore perturbations of the cell membrane in TMEM127-KO cells, we used live-cell time-lapse TIRF imaging of cells stably expressing EGFP-CLC (*Figure 5A*, *Animation 1*). Consistent with our fixed-cell images, Mock-KO cells appeared to have more intense cell surface clathrin puncta, and these moved actively into and out of the TIRF field at the membrane consistent with rapid formation and maturation of CCPs. TMEM127-KO cells had less intense membrane clathrin puncta, which appeared less able to mature as CCPs and internalize into the cell, moving out of the TIRF field.

CCPs undergo a multistage initiation, stabilization, and maturation process (*Mettlen et al., 2018*; *Figure 5B*). Clathrin assemblies must reach a minimum threshold for nucleation of CCPs to initiate. These assemblies can result in either small short-lived (<15 s) abortive CCPs that spontaneously disassemble without cargo uptake, or longer lived productive CCPs that effectively recruit cargo for internalization and delivery to the endolysosomal system (*Kaksonen and Roux, 2018*; *Mettlen et al., 2018*). We examined CCP dynamics over time in live cells using TIRFM and automated detection to identify diffraction-limited EGFP-CLC objects, and assessed the rates of assembly, initiation, and maturation of CCPs in Mock-KO and TMEM127-KO cells. Here, we show that TMEM127-KO cells have significantly increased stochastic assemblies of transient small subthreshold clathrin-labeled structures (sCLS) that are unable to recruit sufficient clathrin to form bona fide CCPs (*Figure 5B and C*). As a result of this increase in transient structures, we show a resultant decrease in the initiation of bona fide CCPs in TMEM127-KO cells compared to Mock-KO cells (*Figure 5B and D*). Further, as indicated by the reduced clathrin puncta intensity at the membrane and reduced area of CCPs detected by dSIM (*Figure 3C and F*), the CCPs that do form in TMEM127-KO cells are smaller, potentially limiting their ability to recruit cargo, such as RET, leading the cargo proteins to accumulate on the cell surface (*Figure 4A–D*). Interestingly, although there are relatively fewer CCPs formed in TMEM127-KO cells (*Figure 5D*), a smaller fraction of these had lifespans less than 15 s, suggesting CCPs that are able to overcome early assembly defects due to TMEM127 loss are able to mature to productive CCPs more efficiently (*Figure 5E*). The overall increase in sCLS and decrease in CCPs suggests that TMEM127 depletion may reduce the potential for clathrin to form or stabilize CCPs capable of internalization and compromise the maturation of productive CCPs, thereby leading to impaired internalization of RET and other cell surface proteins.

The defect in cargo recruitment to CCPs (*Figure 3A and B*) and CCP assembly (*Figures 3C–G and 5A–E*) in TMEM127-KO cells could reflect changes specific to cargo engagement within CCPs or broader changes in membrane dynamics, such as protein and lipid mobility. To determine if loss of TMEM127 resulted in broader impacts on membrane dynamics, we examined the mobility of EGFR by single-particle tracking, using strategies we recently developed and characterized (*Sugiyama et al., 2023*). Importantly, labeling of EGFR with a Fab antibody fragment for single-particle tracking in the absence of ligand stimulation allows study of EGFR mobility that is largely clathrin independent (*Sugiyama et al., 2023*). TMEM127-KO cells exhibited a similar distribution of the population of EGFR in immobile, confined, and mobile cohorts (*Figure 5F*), suggesting that TMEM127 does not disrupt capture of a minor subset of EGFR within tetraspanin nanodomains that occurs in the absence of ligand (*Sugiyama et al., 2023*). However, consistent with broader disruption of membrane dynamics upon loss of TMEM127, we observed an increase in the diffusion coefficient of the confined and mobile fractions, in particular the latter (*Figure 5G*). This predicts that loss of TMEM127 may broadly

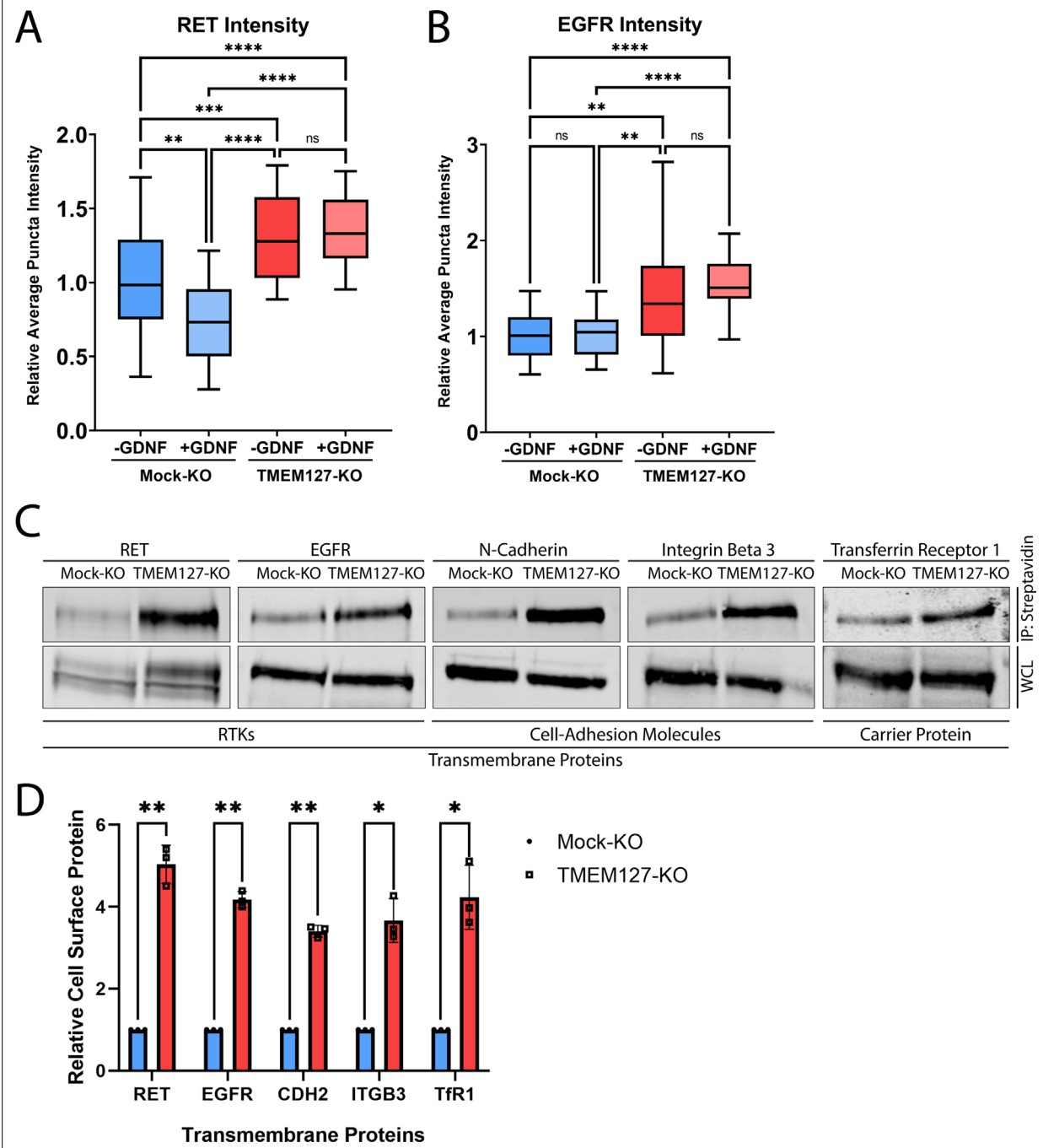

**Figure 4.** TMEM127 depletion leads to cell surface accumulation of transmembrane proteins. (**A**) Total internal reflection fluorescence microscopy (TIRFM) images represented in *Figure 3A* were subjected to automated detection of rearranged during transfection (RET) puncta. Average RET puncta intensity is shown relative to the Mock-KO (-GDNF [glial cell line-derived neurotrophic factor]) condition. Data from n=50 cells representing a minimum of 12,000 RET puncta per condition are shown from four independent experiments (Kruskal-Wallis test and Dunn's multiple comparisons test; **p<0.01, ***p<0.001, ****p<0.0001). (**B**) TIRFM images of Mock-KO and TMEM127-KO SH-SY5Y cells were subjected to automated detection of EGFR puncta. Average EGFR puncta intensity is shown relative to the Mock-KO (-GDNF) condition. Data from n=30 cells representing a minimum of 4600 EGFR puncta per condition are shown from two independent experiments (Brown-Forsythe and Welch ANOVA tests and Dunnett's T3 multiple comparisons test; **p<0.01, ****p<0.0001). (**C**) Immunoblot showing the indicated total and cell surface biotinylated transmembrane proteins in Mock-KO and TMEM127-KO SH-SY5Y cells. Biotinylated proteins include RET, EGFR, N-cadherin (CDH2), integrin beta-3 (ITGB3), and transferrin receptor-1 (TfR1). Biotinylated proteins were collected, separated, and immunoblotted as in *Figure 2A*. (**D**) Quantification of relative cell surface protein levels detected in **C**. Biotinylated cell surface protein levels (immunoprecipitation [IP]) were normalized to corresponding total protein (whole cell lysate [WCL]) and

*Figure 4 continued on next page*

*Figure 4 continued*

expressed relative to Mock-KO for each protein. Three independent experiments (n=3) are shown as mean ± SD (two-tailed unpaired t-tests with Welch's correction; *p<0.05, **p<0.01).

The online version of this article includes the following source data for figure 4:

**Source data 1.** Excel file of rearranged during transfection (RET) and EGFR total internal reflection fluorescence (TIRF) intensity and immunoblot quantification analyses shown in *Figure 4*.

**Source data 2.** Original files for immunoblot analysis in *Figure 4C*.

**Source data 3.** Labeled file for immunoblot analysis shown in *Figure 4C*.

impact membrane dynamics, suggesting that defects in receptor recruitment to CCPs and perturbations of CCP assembly that may lead to accumulation of membrane proteins at the cell surface in TMEM127-KO cells may result in part from broad alterations in membrane dynamics and/or receptor mobility that impact the formation of CCPs and their recruitment of diverse cargo.

## Clathrin adaptor recruitment is not TMEM127 dependent

Recruitment of RET to CCPs requires interaction with the clathrin adaptor AP2 (*Crupi et al., 2015*). We used proximity ligation assays (PLA) to identify and quantify direct protein interactions in situ to assess association of endogenous AP2 with RET and clathrin in SH-SY5Y cells. In untreated Mock-KO cells, RET-AP2 interaction was minimal but was significantly increased by GDNF treatment, as indicated by increased PLA puncta (*Figure 6A*), suggesting RET recruitment as cargo to CCPs. Association of RET and AP2 was significantly greater in TMEM127-KO cells, consistent with the higher RET protein levels at the cell membrane, and was further increased by GDNF treatment, suggesting that the RET-AP2 association was not TMEM127 dependent. We saw robust association of clathrin and AP2 in untreated Mock-KO cells (*Figure 6B*). GDNF treatment significantly reduced association, consistent with maturation of productive CCPs and subsequent uncoating of clathrin-coated vesicles and delivery of cargo to endosomes. In comparison, TMEM127-KO cells had significantly greater constitutive association of AP2 and clathrin and this was not affected by GDNF. We did not detect association of AP2 with TMEM127 under any condition using PLA (not shown), suggesting TMEM127 is not closely associated with clathrin-labeled structures. Together, our data suggest that, in TMEM127-KO cells, AP2-clathrin complexes accumulate but are unable to efficiently support the formation and/or assembly of CCPs to facilitate internalization of cargo proteins. This is consistent with the observation that TMEM127-KO cells have significantly higher sCLS nucleation rates, while exhibiting decreased formation of bona fide CCPs (*Figure 5C and D*). It follows that the increased proximity of RET and AP2 in TMEM127-KO cells may reflect increased cell surface levels of RET that may engage with endocytic adaptors, but that these interactions infrequently lead to formation or stabilization of bona fide CCPs. These data suggest that TMEM127 depletion does not directly affect ability of adaptors to associate with clathrin or cargo but may reflect an integral membrane defect that alters normal membrane protein movement and abilities to assemble or stabilize larger membrane protein complexes.

## TMEM127 depletion alters membrane microdomain organization

Given the results of the EGFR single-particle tracking experiments that suggest broader disruption of membrane dynamics upon loss of TMEM127 (*Figure 5F and G*), we investigated whether loss of TMEM127 could more broadly impact membrane organization by assessing distribution of lipid-rich membrane microdomains (membrane rafts). In Mock-KO cells stained for ganglioside $G_{M1}$, which selectively partitions into membrane rafts (*Sonnino et al., 2007*), stained lipid microdomains were large and continuous (*Figure 6C*). In TMEM127-KO cells, staining was fragmented and microdomains were significantly smaller, suggesting inability to assemble large lipid clusters and raft-associated proteins in the surface membrane. These data suggest that TMEM127-depletion alters membrane dynamics, limiting formation of multiple membrane structures.

## Decreased internalization reduces RET degradation, increasing RET half-life

Internalization of activated RET into the endolysosomal system is required for downregulation of RET signaling and eventual degradation (*Crupi et al., 2020*; *Hyndman et al., 2017*). As RET internalization

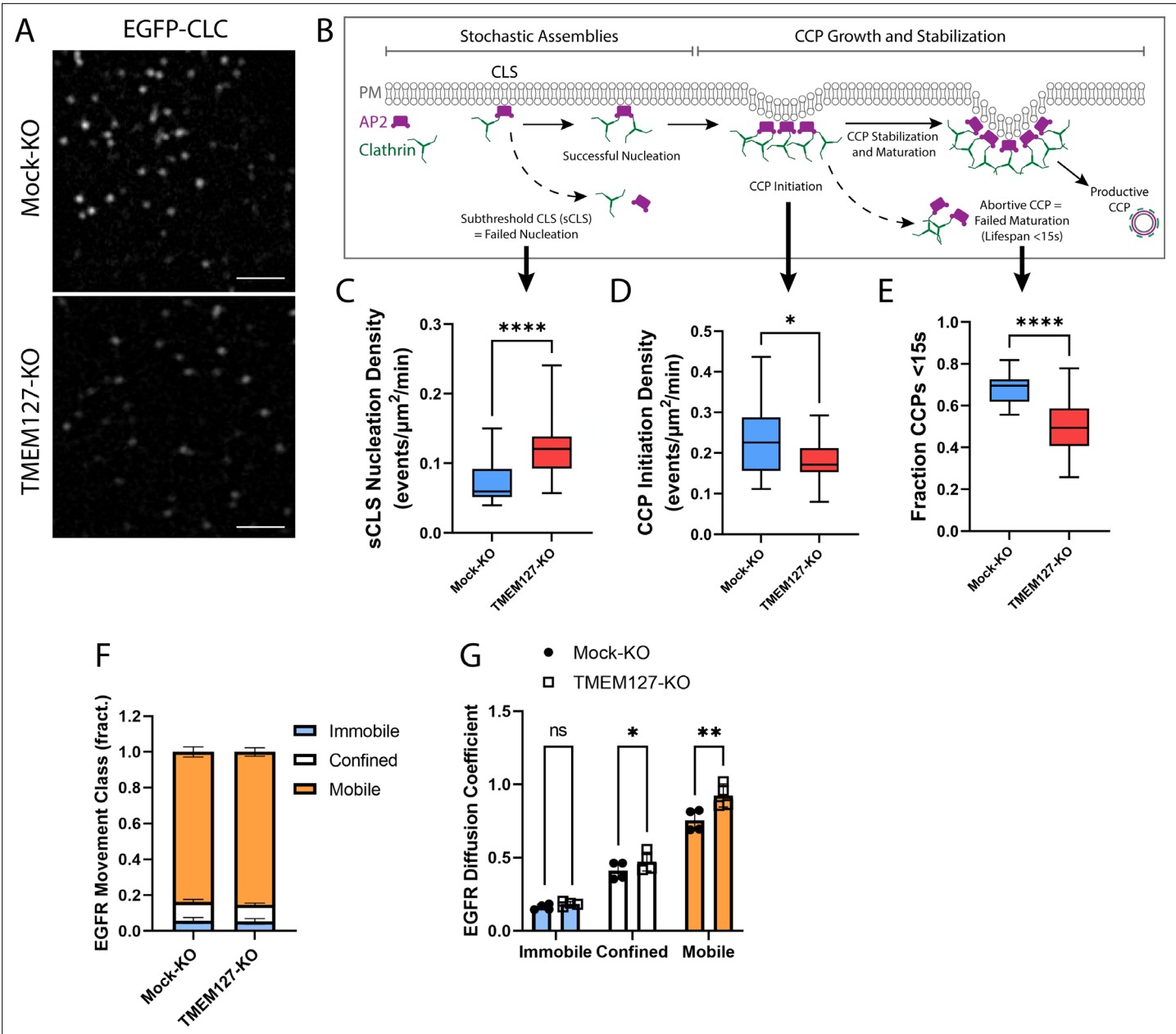

**Figure 5.** TMEM127 loss alters dynamics of clathrin-coated pit (CCP) assembly and EGFR diffusion. Live-cell total internal reflection fluorescence microscopy (TIRFM) imaging of cell surface EGFP-tagged clathrin light chain (EGFP-CLC) stably expressed in Mock-KO and TMEM127-KO SH-SY5Y cells. (**A**) Representative TIRFM images of cell surface clathrin distribution. Scale bars = 2 µm. (**B**) Diagram of the stepwise process of CCP assembly. Plasma membrane (PM) (gray), AP2 (purple), and clathrin (green) are indicated. Stochastic assemblies of clathrin adaptor AP2 and clathrin form clathrin-labeled structures (CLS). Small structures with insufficient clathrin are unstable and may be unable to nucleate clathrin polymerization (subthreshold CLS [sCLS]) and dissipate or may successfully stimulate clathrin polymerization, leading to CCP initiation. As polymerization continues, the CCP undergoes maturation and stabilization processes required to develop a productive CCP, which functions to internalize protein cargo into the cell. Unsuccessful CCP maturation leads to abortive CCPs with lifespans <15 s. Steps relevant to graphs C–E are indicated with black arrows. (**C–E**) Time-lapse videos of cells (5 min @ 1 frame-per-second) were subjected to automated detection of cell membrane clathrin structures, which included sCLS and bona fide CCPs. (**C**) sCLS nucleation density, (**D**) CCP initiation density, and (**E**) fraction CCPs <15 s are shown. (**C–E**) The means of multiple time-lapse videos are shown. In three independent experiments, the number of total sCLS trajectories, CCP trajectories, and time-lapse videos (respectively) for each condition are Mock-KO, 11040, 7988, 20, and TMEM127-KO, 15598, 8393, 28 (C–D: two-tailed Mann-Whitney test, E: two-tailed unpaired t-test with Welch's correction; *p<0.05, ****p<0.0001). (**F**) EGFR movement class shown as the mean ± SD fraction (fract.) of all EGFR tracks, as labeled by Fab-Cy3B, that exhibit mobile (orange), confined (white), or immobile (blue) behavior. (**G**) EGFR diffusion coefficient shown is mean ± SD of immobile, confined, and mobile EGFR tracks. (**F–G**) Data from four independent experiments representing detection and tracking of minimum 500 EGFR objects (two-tailed paired t-tests; *p<0.05, **p<0.01).

*Figure 5 continued on next page*

*Figure 5 continued*

The online version of this article includes the following source data for figure 5:

**Source data 1.** Excel file of total internal reflection fluorescence (TIRF) data and EGFR movement quantification analyses shown in *Figure 5*.

was reduced in TMEM127-KO cells, we assessed whether this might impair RET degradation by examining RET protein expression levels over time following GDNF treatment in cells treated with cycloheximide (CHX) to inhibit new protein synthesis. In GDNF-treated Mock-KO cells, RET was rapidly degraded, with a half-life under 2 hr and less than 9% RET protein remaining after 12 hr (*Figure 7*). In these cells, TMEM127 was shown to have a half-life of approximately 5 hr. In contrast, RET degradation was significantly slower in TMEM127-KO cells, with a half-life of approximately 6 hr and over 43% of RET remaining after 12 hr (*Figure 7*). These data suggest that impaired RET internalization due to altered membrane dynamics significantly reduces RET degradation, contributing to RET accumulation on the cell surface.

## RET accumulation promotes increased downstream signaling

Overexpression or cell surface accumulation can frequently cause constitutive activation of growth factor receptors, conferring ligand insensitivity for modulating the intensity and duration of their downstream signals (*Du and Lovly, 2018*). We assessed RET activation and signaling over time in Mock-KO and TMEM127-KO cells in response to GDNF. In Mock-KO cells, RET and its downstream signaling intermediates AKT, ERK, and the mTOR substrate S6 were minimally phosphorylated in the absence of GDNF and showed significantly increased phosphorylation over time in response to GDNF treatment (*Figure 8A and B*), consistent with our previous studies (*Crupi et al., 2020*; *Richardson et al., 2006*). In contrast, RET, AKT, ERK, and S6 were constitutively phosphorylated in TMEM127-KO cells. GDNF enhanced RET phosphorylation after 5 min but, unlike Mock-KO cells, the effect was not further increased by prolonged treatment times. Phosphorylation of AKT, ERK, and S6 was not significantly increased by GDNF treatment in TMEM127-KO cells (*Figure 8A and B*). Further, we showed that the constitutive phosphorylation of RET, AKT, ERK, and S6 in TMEM127-KO cells could be blocked by RET inhibition using the multikinase inhibitor vandetanib, and selective RET inhibitors selpercatinib and pralsetinib (*Figure 8C*), which are clinically approved for the treatment of RET-associated cancers (*Mulligan, 2018*). GDNF-mediated phosphorylation in Mock-KO cells was also blocked by RET inhibition (*Figure 8—figure supplement 1*). Together, our data show that cell surface accumulation of RET can promote constitutive phosphorylation and activation of multiple downstream signaling pathways.

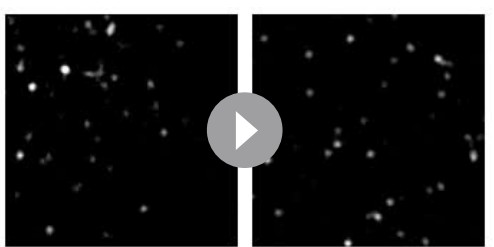

**Animation 1.** Assembly of clathrin-labelled structures is reduced in the absence of TMEM127 – Live cell TIRFM imaging movies of cell surface clathrin in Mock-KO and TMEM127-KO SH-SY5Y cells expressing EGFP-tagged clathrin light chain. Time-lapse videos of cells (5 min @ 1 frame-per-second) are shown. (Left) In Mock-KO cells clathrin-labeled structures (CLS) assemble rapidly and abundantly. These may abort (very short-lived puncta) or progress to mature clathrin-coated pit (CCP) before internalizing (~40 s) and disappearing from the TIRF field (*Crupi et al., 2015*) (Right) TMEM127-KO cells show fewer CLS, which assemble more slowly and remain on the membrane much longer before moving out of the TIRF field, suggesting reduced recruitment of clathrin to CLS in these cells, contributing to reduced ability of CCPs to mature and capture protein cargo.

## RET drives proliferation in TMEM127-depleted cells

Oncogenic activating RET mutations are drivers of proliferation and tumor growth in PCC (*Mulligan, 2018*; *Neumann et al., 2019*; *Toledo et al., 2017*). Here, we assessed whether RET constitutive activation caused by cell surface accumulation in response to TMEM127 loss could also promote these processes. We showed that GDNF treatment significantly increased proliferation of Mock-KO cells over a 3-day period (*Figure 8D*). TMEM127-KO cells proliferated significantly faster than Mock-KO cells, irrespective of GDNF treatment (*Figure 8D*), consistent with increased RET accumulation leading to ligand-independent RET activity and proliferation. To confirm that proliferation was RET-mediated and explore the impact of RET inhibition in TMEM127-KO cells,

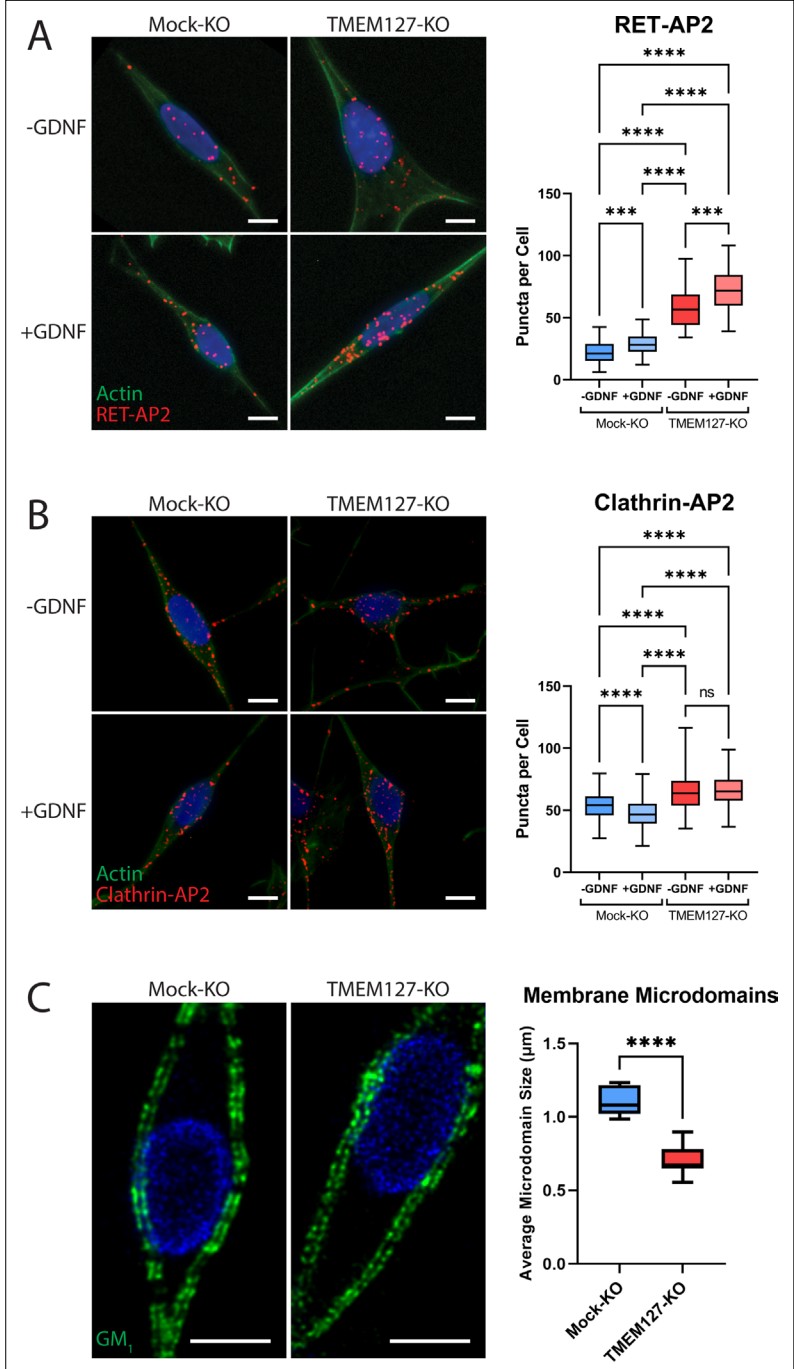

**Figure 6.** Enhanced recruitment of endocytic adaptors and disrupted membrane microdomains in TMEM127-depleted cells. (**A and B**) Proximity ligation assay (PLA) of Mock-KO and TMEM127-KO SH-SY5Y cells treated with (+) or without (-) glial cell line-derived neurotrophic factor (GDNF) (100 ng/ml) for 5 min. Representative immunofluorescence images (left) of RET-AP2 (**A**) and clathrin-AP2 (**B**) interactions (red puncta). Cells were stained with phalloidin actin filament stain (green) and Hoechst nuclear stain (blue). Quantification of red puncta per cell (right), indicating individual protein interactions in each condition, are shown. Data from a minimum of 100 cells (-GDNF conditions) (**A**), 300 cells (+GDNF conditions) (**A**), or 500 cells (all conditions) (**B**) are shown (Kruskal-Wallis test and Dunn's multiple comparisons test; ***p<0.001, ****p<0.0001). Scale bars = 5 µm. (**C**) Immunofluorescence confocal images of Mock-KO and TMEM127-KO SH-SY5Y cells (left) stained with cholera toxin subunit B, which binds ganglioside $GM_1$-positive lipid microdomains (membrane rafts) (green), and Hoechst nuclear stain (blue). Quantification of average lipid raft size (right) of n=11–12 cells per condition are shown, representing a minimum of 500 lipid rafts per condition (two-tailed unpaired t-test; ****p<0.0001). Scale bars = 5 µm.

*Figure 6 continued on next page*

*Figure 6 continued*

The online version of this article includes the following source data for figure 6:

**Source data 1.** Excel file of proximity ligation assay (PLA) data and membrane microdomain quantification analyses shown in *Figure 6*.

we assessed proliferation in the presence and absence of the selective RET inhibitor selpercatinib at a concentration sufficient to block RET phosphorylation (0.1 μM, *Figure 8—figure supplement 2*). Selpercatinib significantly reduced GDNF-dependent proliferation of Mock-KO cells. In TMEM127-KO cells, RET inhibition abrogated the increase in proliferation, seen above, and further significantly reduced viability (~20%) compared to Mock-KO cells (*Figure 8E*). Vehicle alone (DMSO) did not affect proliferation under any conditions (*Figure 8—figure supplement 3*). Our data demonstrate that proliferation of TMEM127-KO cells was RET-dependent and suggest a previously unrecognized sensitivity of TMEM127-depleted cells to RET inhibition.

## Discussion

The RET receptor and the TMEM127 integral membrane protein are established PCC susceptibility genes (*Dahia, 2014*; *Mulligan, 2018*; *Neumann et al., 2019*). RET oncogenic mutations give rise to PCC in patients with MEN2 and are found in up to 5% of sporadic PCC, where they promote constitutive RET activation and downstream signaling (*Horton et al., 2022*; *Le Hir et al., 2000*; *Mulligan, 2018*; *Takaya et al., 1996a*; *Takaya et al., 1996b*). In contrast, TMEM127 acts as a tumor suppressor, and loss-of-function mutations have been linked primarily to increased mTOR signaling (*Deng et al., 2018*). Although RET and WT-TMEM127 localize to similar endosomal compartments, we show that colocalization is lost in the presence of PCC-associated mutants that disrupt TMEM127 membrane insertion (*Qin et al., 2010*). Endosomes are reduced in size and number in the absence of WT-TMEM127, suggesting defects in the processes required for vesicle formation and internalization and trafficking of proteins from the plasma membrane (*Figure 9*).

Using a CRISPR TMEM127-KO in the neuroblastoma cell line SH-SY5Y, we noted significantly increased levels of endogenously expressed RET protein, which appeared primarily as the higher molecular weight fully glycosylated form found at the cell surface (*Richardson et al., 2006*; *Richardson*

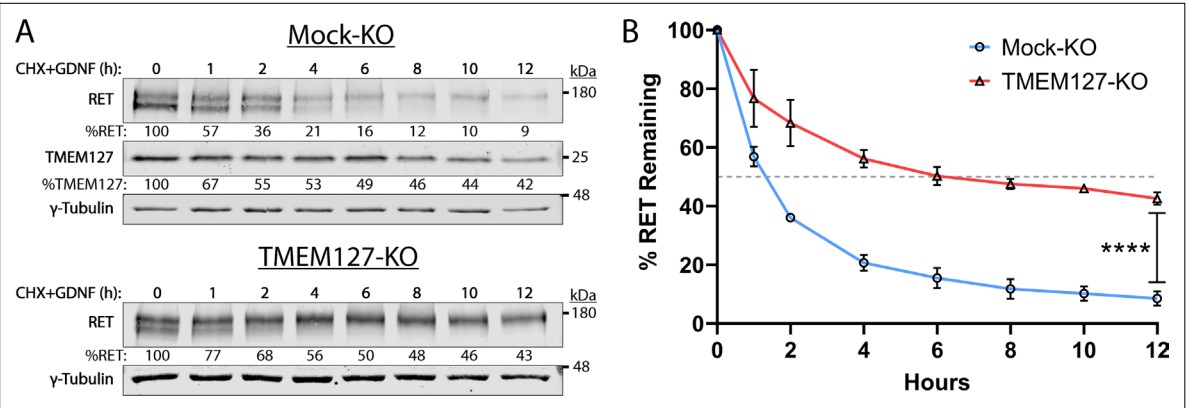

**Figure 7.** Rearranged during transfection (RET) half-life is increased when TMEM127 is depleted. (**A**) Mock-KO (upper panels) and TMEM127-KO (lower panels) SH-SY5Y cells were treated with cycloheximide (CHX) and glial cell line-derived neurotrophic factor (GDNF) (100 ng/ml) for indicated times to evaluate RET and TMEM127 protein levels remaining over time. Whole cell lysates were separated by SDS-PAGE and immunoblotted for the indicated proteins. γ-Tubulin was used as a loading control. RET and TMEM127 protein remaining at each timepoint, relative to untreated levels (0 h), is indicated below each lane. (**B**) Quantification of RET protein remaining over time as shown in (**A**). The dashed line indicates 50% RET remaining. The means ± SD at each timepoint of three independent experiments (n=3) are shown (two-way ANOVA with Šídák's multiple comparisons test; ****p<0.0001).

The online version of this article includes the following source data for figure 7:

**Source data 1.** Excel file of immunoblot quantification data shown in *Figure 7B*.

**Source data 2.** Original files for immunoblot analysis in *Figure 7A*.

**Source data 3.** Labeled file for immunoblot analysis shown in *Figure 7A*.

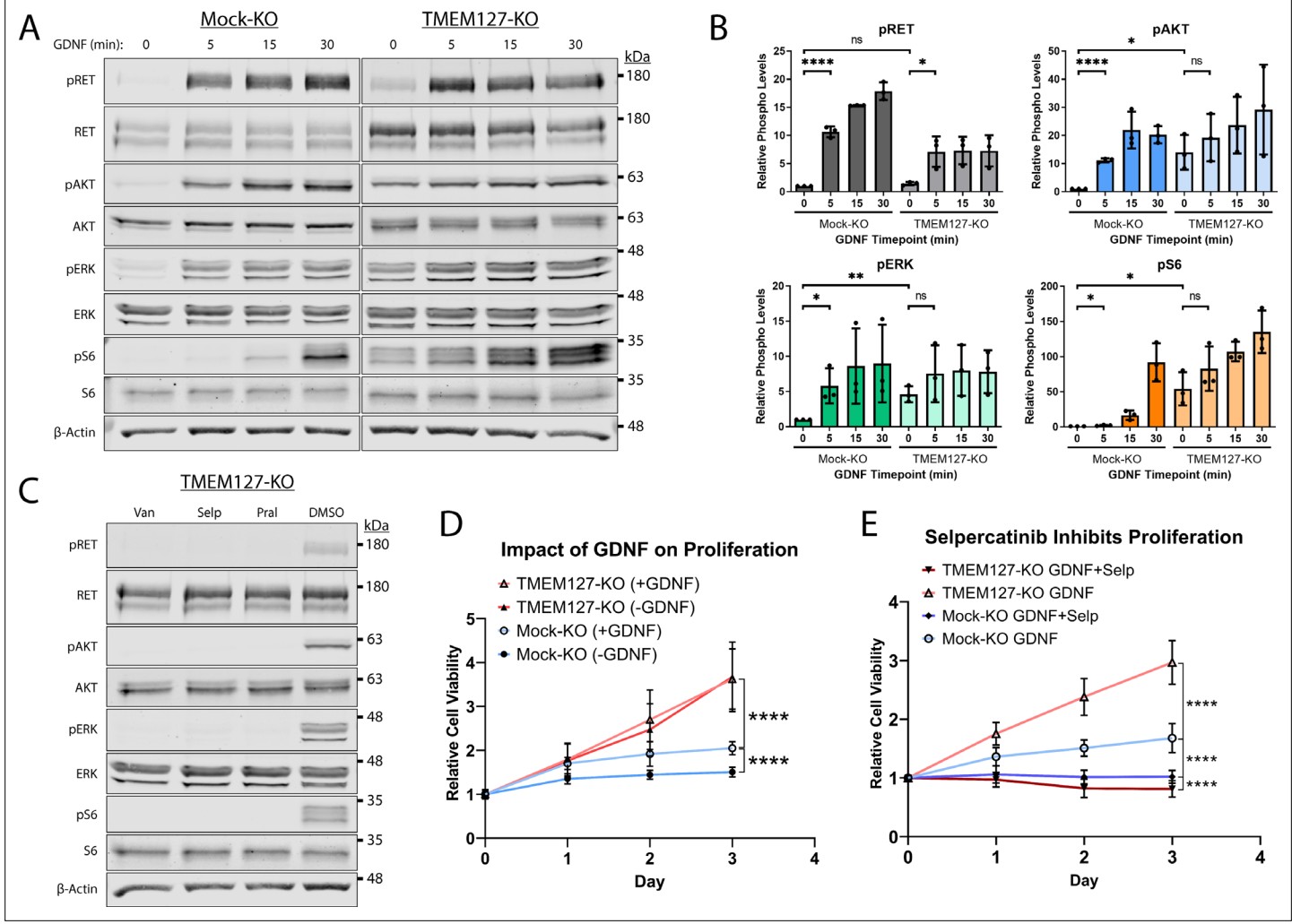

**Figure 8.** Constitutive rearranged during transfection (RET)-mediated signaling drives proliferation and confers sensitivity to selpercatinib. (**A**) Mock-KO and TMEM127-KO SH-SY5Y cells were serum starved and treated with glial cell line-derived neurotrophic factor (GDNF) for the indicated times. Whole cell lysates were separated by SDS-PAGE and immunoblotted for the indicated phospho and total proteins. β-Actin was used as a loading control. (**B**) Quantification of phosphorylated protein levels as shown in **A** relative to Mock-KO 0 min condition for each protein. The means ± SD of three independent experiments (n=3) are shown (two-tailed unpaired t-tests; *p<0.05, **p<0.01, ****p<0.0001). (**C**) Expression of the indicated phospho and total proteins in TMEM127-KO SH-SY5Y cells treated with DMSO, 5 µM vandetanib (Van), selpercatinib (Selp), or pralsetinib (Pral) (1 hr), and immunoblotted as in (**A**). Constitutive phosphorylation of indicated proteins was abrogated by all three inhibitors. (**D**) Cell viability of Mock-KO and TMEM127-KO SH-SY5Y cells over time in the presence or absence of GDNF, relative to day 0 was measured by MTT assay as an indicator of cell proliferation. (**E**) Cell viability of Mock-KO and TMEM127-KO SH-SY5Y cells in the presence of GDNF (100 ng/ml) alone or GDNF+selpercatinib (0.1 µM), relative to day 0, as above. (**D and E**) Three independent experiments, for each graph, with 18 replicates per condition and timepoint (n=54) are shown as mean ± SD (two-way ANOVA with Tukey's multiple comparisons test; ****p<0.0001).

The online version of this article includes the following source data and figure supplement(s) for figure 8:

**Source data 1.** Excel file of viability assay and immunoblot quantification analyses shown in *Figure 8B, D, and E*.

**Figure supplement 1.** Rearranged during transfection (RET) activation promoted AKT, ERK, and S6 phosphorylation in SH-SY5Y cells.

**Figure supplement 1—source data 1.** Original files for immunoblot analyses shown in *Figure 8A and C* and *Figure 8—figure supplement 1*.

**Figure supplement 1—source data 2.** Labeled file for immunoblot analyses shown in *Figure 8A and C* and *Figure 8—figure supplement 1*.

**Figure supplement 2.** Optimization of selpercatinib concentration.

**Figure supplement 2—source data 1.** Original files for immunoblot analyses in *Figure 8—figure supplement 2*.

**Figure supplement 2—source data 2.** Labeled file for immunoblot analyses shown in *Figure 8—figure supplement 2*.

**Figure supplement 3.** Vehicle treatment does not significantly affect SH-SY5Y cell proliferation.

**Figure supplement 3—source data 1.** Excel file of viability assay vehicle controls shown in *Figure 8—figure supplement 3*.

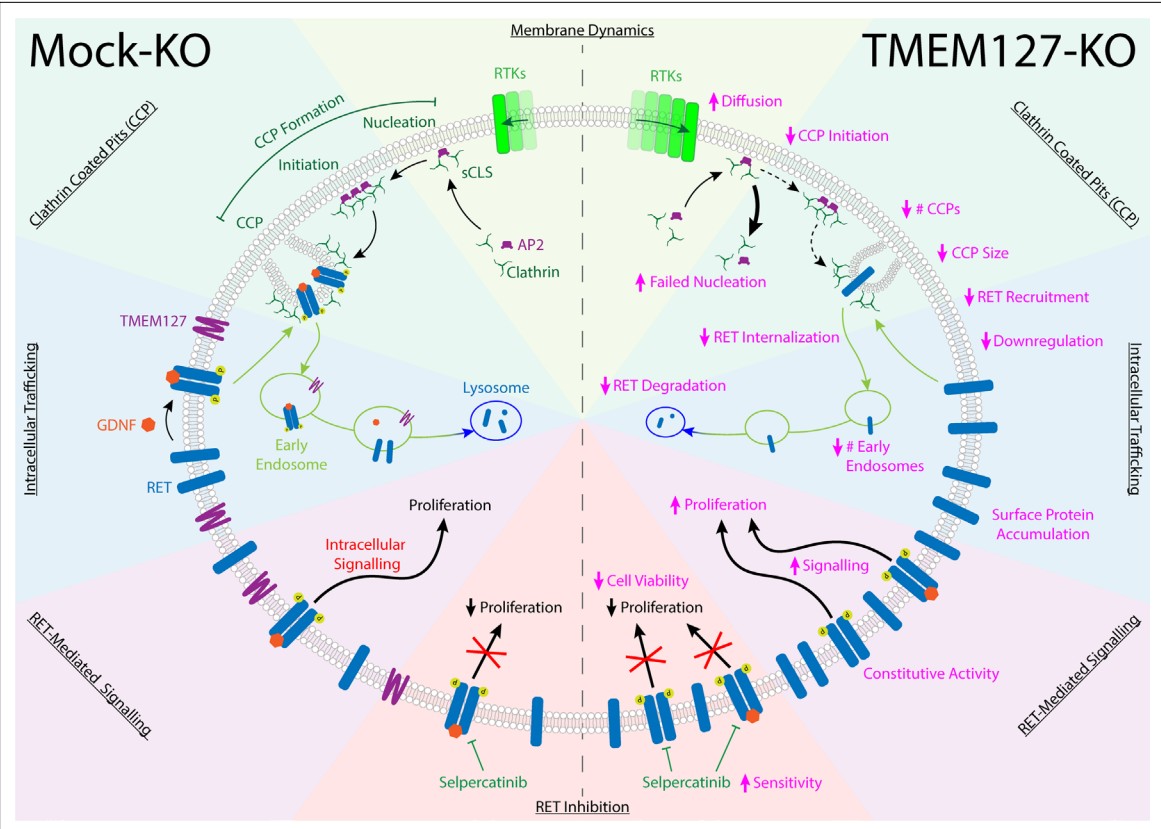

**Figure 9.** TMEM127 loss drives transformation through altered rearranged during transfection (RET) location and activity in pheochromocytoma (PCC). TMEM127 depletion alters membrane diffusion and impairs formation of productive clathrin-coated pits (CCPs) for internalization of membrane receptors. CCPs are fewer, smaller, and less able to recruit cargo, resulting in accumulation of transmembrane proteins, including RET on the cell surface. As a result, RET downstream signaling is enhanced and degradation is reduced leading to increased cell proliferation. The selective RET inhibitor selpercatinib abrogates RET-mediated signals and blocks cell proliferation.

*et al., 2012*; *Takahashi et al., 1991*). Using surface biotin labeling and TIRFM in fixed and live cells, we showed that RET internalization was impaired, and that WT RET accumulated on the cell surface in the absence of TMEM127, escaping degradation (*Figure 9*). Notably, accumulation of surface receptors was not unique to RET as we also saw increased intensity of EGFR puncta and increased membrane levels of other receptors and adhesion molecules in the absence of TMEM127 in both SH-SY5Y and HEK293 cells, consistent with a global defect of membrane dynamics that could also block internalization of membrane proteins. As a result of increased density in the plasma membrane in SH-SY5Y cells, RET is constitutively active, promoting stimulation of multiple downstream signaling pathways. Importantly, treatment with selective RET inhibitors abrogated these signals including mTOR signaling, suggesting that, in this model, the increase in mTOR signaling seen in response to TMEM127 loss is a direct response to RET activity.

As both RET and TMEM127 have been previously shown to internalize through CME (*Crupi et al., 2020*; *Crupi et al., 2015*; *Flores et al., 2020*; *Richardson et al., 2006*), we assessed the impact of TMEM127 depletion on membrane clathrin distribution and its effects on cargo internalization. Despite increased RET puncta intensity on the cell surface, RET localization to clathrin puncta was reduced and was not responsive to GDNF, consistent with impairment of clathrin-mediated internalization of RET (*Figure 9*). The formation of CCPs is a multistep process involving recruitment of clathrin adaptors and stepwise assembly of clathrin-labeled structures at the cell membrane for recruitment of cargo proteins and assembly of productive CCPs (*Mettlen et al., 2018*). Transient small subthreshold CLSs are unstable and may not recruit sufficient clathrin or cargo to achieve a threshold size and progress further to initiate bona fide CCPs (*Mettlen et al., 2018*). Once sufficient clathrin is recruited for initiation, CCPs may mature to pinch off clathrin-coated vesicles from the membrane, but a subset fail to mature and are aborted (*Kadlecova et al., 2017*; *Mettlen et al., 2018*). Our single-particle tracking

studies suggest that mobility of membrane cargo proteins like EGFR is enhanced in TMEM127 KO cells, which may decrease capture of cargo at stochastic clathrin assemblies and limit their progression to successful nucleation and initiation of bona fide CCPs. Consistent with this, our data indicate that TMEM127 depletion increased the frequency of the sCLSs but significantly reduced their ability to reach the clathrin threshold size and initiate bona fide CCPs. CCPs that did form were smaller, contained less clathrin, and were able to recruit less RET cargo than in Mock-KO cells. TMEM127 loss did not impair RET-AP2 or AP2-clathrin association, suggesting that key protein interactions required for internalization were largely intact (*von Zastrow and Sorkin, 2021*). However, we saw decreased capture of RET within bona fide CCPs, formation of CCPs was reduced, and those that formed smaller in TMEM127-deficient cells, suggesting that altered membrane dynamics, increasing the diffusability of membrane proteins and impairing capture of sufficient cargo-adaptor complexes into nascent CCPs, could be limiting the recruitment of sufficient cargo and coalescence of CCPs, consistent with previous studies that link membrane tension and CCP formation (*Djakbarova et al., 2021*; *Saleem et al., 2015*). Together, our data suggest that accumulation of RET and other transmembrane proteins at the cell surface in the absence of functional TMEM127 is due to reduced efficiency of clathrin recruitment to CCP, limiting capacity for cargo internalization and reducing CME, and that this correlates with the global disruption of membrane protein composition, protein complex assembly, and organization in other domains, such as membrane rafts. Interestingly, our preliminary studies have suggested that one outcome of this may be impairment of the formation of other membrane-associated complexes and processes, such as recruitment of ubiquitination machinery (*Guo et al., 2023*), contributing to the TMEM127 phenotype and highlighting the broader implications of altered membrane partitioning.

As a result of impaired internalization in TMEM127-deficient SH-SY5Y cells, RET protein accumulates on the cell surface and degradation is reduced, leading to sustained downstream proproliferative signals. We showed that these pathways can be blocked using selective RET inhibitors, suggesting that targeting RET may be a valuable therapeutic strategy in the subset of PCC tumors where RET mutations have not been recognized but RET expression is detected. In support of this, in recent preliminary studies, we have shown that xenografts of TMEM127 KO SH-SY5Y cells produce larger tumors in nude mice, but that growth is reduced by treatment with the RET inhibitor selpercatinib (*Guo et al., 2023*). Interestingly, TMEM127 mutations have also been identified as drivers in other cancers, notably renal cell carcinoma, where RET is not highly expressed. In these cancers, we would predict that TMEM127 loss has similar effects on membrane dynamics, membrane protein accumulation, and CCP formation to those seen in our model and that accumulation of other growth factor receptors, such as MET, on the cell surface will lead to transformation (*Marona et al., 2019*; *Rhoades Smith and Bilen, 2019*). Notably, TMEM127 mutations or reduced expression are also seen at low frequencies in a number of other tumor types (e.g. endometrial, liver, breast, kidney, ovarian cancers; *Tate et al., 2019*) where TMEM127 contributions have not yet been specifically recognized, suggesting that aberrant accumulation and activation of growth factor receptors could also impact these tumors. Together, our data suggest a novel paradigm for oncogenic transformation in TMEM127-mutant tumors. We predict that altered membrane dynamics blocks normal internalization and degradation of key WT growth-promoting receptors, which then act as oncogenes to drive tumorigenesis in a cell type-specific fashion. Further, we predict that these oncogenes will provide valuable alternative therapeutic targets that have not been previously explored in TMEM127-mutant cancers.

Importantly, the changes in membrane dynamics and aberrant accumulation of membrane proteins in response to TMEM127 depletion are not exclusively oncogenic but may have other pathological implications. Consistent with our data showing RET accumulation on the cell surface, TMEM127 KO mice have increased insulin sensitivity and upregulation of insulin-dependent AKT signaling (*Srikantan et al., 2019*), which may suggest excess surface accumulation of insulin receptors and resulting enhanced signaling in the liver in this model. Notably, recent studies have suggested that TMEM127 may have immune modulatory roles. In *Salmonella*-infected but not normal antigen presenting cells, TMEM127 acts in recruitment of protein complexes that promote downregulation of MHC-II and suppression of T-cell activation, while depletion of TMEM127 in infected cells increases MHC-II surface expression and reduces innate immune responses (*Alix et al., 2020*). Similarly, TMEM127 depletion leads to MHC-I accumulation on the cell membrane in AML cell models, promoting enhanced recruitment of CD8[+] T cells (*Chen et al., 2023*). Taken together, these data suggest that the outcomes of TMEM127 depletion may be cell type as well as tumor type and stage specific, depending on the cell

membrane proteomic repertoire. Further, TMEM127 depletion may promote oncogenic signaling in tumor initiation and growth, as we observed in PCC, or potentially contribute to alterations in the tumor immune environment in progression of other cancer models. These observations suggest that TMEM127 may have broad cellular roles in maintaining balance in the membrane proteome and suggest a generalized role for TMEM127 as a facilitator of diverse membrane protein complexes regulating localized membrane diffusion and promoting stability of complex assemblies to regulate membrane organization.

## Materials and methods

### Cell culture

Human Retinal Pigment Epithelial ARPE-19 cells (CRL-2302, ATCC, Manassas, VA, USA, RRID:CVCL_0145) were cultured in Ham's F-12 medium (Thermo Fisher Scientific, Waltham, MA, USA) with 10% fetal bovine serum (FBS; Sigma-Aldrich, Oakville, ON, Canada). TMEM127-KO and Mock-KO were generated in SH-SY5Y neuroblastoma cells (CRL-2266, ATCC, RRID:CVCL_0019) and HEK293 (CRL-1573, ATCC, RRID:CVCL_0045) using CRISPR-Cas9 as previously described (*Deng et al., 2018*). SH-SY5Y and HEK293 cells were maintained in Dulbecco's Modified Eagle's Medium (DMEM; Sigma-Aldrich) with 10% and 5% FBS respectively, 10 µg/ml ciprofloxacin, and 2.5 µg/ml puromycin. Cell lines were authenticated by STR profiling (Centre for Applied Genomics, SickKids, Toronto, ON, Canada). EGFP-CLC expressing variants of these cells were generated through lentiviral transduction, as previously described (*Crupi et al., 2020*), and selected in 100 µg/ml Hygromycin B (BioShop, Burlington, ON, Canada). Prior to each experiment, endogenous RET expression in SH-SY5Y was stimulated with 5 µM retinoic acid (MilliporeSigma, Oakville, ON, Canada) for 12–18 hr. Where appropriate, RET activation was induced with 100 ng/ml GDNF (Alomone Labs, Israel) for the indicated times.

### Antibodies, stains, and inhibitors

Total RET (#14556, RRID:AB_2798509), pRET (#3221, RRID:AB_2179887), AKT (#9272, RRID:AB_329827), pAKT (4058, RRID:AB_331168), S6 (#2317, RRID:AB_2238583), and pS6 (#2211, RRID:AB_331679) antibodies were from Cell Signaling Technologies (Beverly, MA, USA). Antibodies to RET51 (#sc-1290, RRID:AB_631316), β-actin (#sc-47778, RRID:AB_626632), EGFR (#sc-120, RRID:AB_627492), ERK1 (#sc-94, RRID:AB_2140110), pERK (#sc-7383, RRID:AB_627545), EGFP (#sc-9996, RRID:AB_627695), and EEA1 (#sc-137130, RRID:AB_2246349) were from Santa Cruz Biotechnology (Dallas, TX, USA). The anti-γ-tubulin (#T6557, RRID:AB_477584) antibody was from Sigma-Aldrich. Clathrin Heavy Chain antibodies (#ab2731 [RRID:AB_303256] and #ab21679 [RRID:AB_2083165]), RET (# ab134100, RRID:AB_2920824), and alpha-adaptin (ab2730, RRID:AB_303255) were from Abcam (Cambridge, UK). The TMEM127 (#A303-450A, RRID:AB_10952702) antibody was from Bethyl Laboratories (Fortis Life Sciences, Waltham, MA, USA). Western blot IRDye secondary antibodies were from LI-COR Biosciences (Lincoln, NE, USA). Alexa Fluor-conjugated secondary antibodies (Alexa-488, -546, -594, and -647) and Phalloidin (Alexa-488) were from Thermo Fisher Scientific. Cyanine Cy3- and Cy5-conjugated AffiniPure fluorescent secondary antibodies were from Jackson ImmunoResearch Laboratories Inc (West Grove, PA, USA). Nuclear stains used include Hoechst 33342 (Thermo Fisher Scientific) and DAPI 10236276001 (Roche Diagnostics, Mannheim, Germany). Kinase inhibitors used were vandetanib (AstraZeneca PLC, Mississauga, ON, Canada), selpercatinib, and pralsetinib (Selleckchem, Houston, TX, USA).

### Expression constructs and transfection

Expression constructs for WT RET51, GFRα1, and FLAG- or EGFP-tagged TMEM127 WT and mutants (C140Y and S147del) have been previously described (*Hyndman et al., 2017*; *Qin et al., 2010*; *Richardson et al., 2012*). EGFP-tagged rat clathrin light chain (*Ehrlich et al., 2004*) was cloned into pLenti CMV and sequence verified.

Cells were transiently transfected with the indicated constructs using Lipofectamine 2000 (Invitrogen, Burlington, ON, Canada), according to the manufacturer's instructions (*Crupi et al., 2020*). For ARPE-19 cells, medium was changed 6 hr after transfection and cells were incubated at 37°C for 48 hr prior to fixation. For SH-SY5Y and HEK293 cells, medium was changed 5 hr after transfection and cells were incubated at 37°C overnight (16 hr) prior to harvesting.

## Protein isolation and immunoblotting

Cells were washed with phosphate-buffered saline (PBS) and protein was harvested with lysing buffer (20 mM Tris-HCl [pH 7.8], 150 mM NaCl, 1 mM sodium orthovanadate, 1% Igepal, 2 mM EDTA, 1 mM phenylmethylsulfonyl fluoride, 10 µg/ml aprotinin, and 10 µg/ml leupeptin), as described (*Crupi et al., 2020*; *Hyndman et al., 2017*). Proteins were separated by SDS-PAGE and transferred to nitrocellulose membranes, as previously described (*Crupi et al., 2020*; *Hyndman et al., 2017*). Blots were probed with indicated primary (1:1000–1:5000 dilution) and secondary IRDye antibodies (1:20,000) and visualized using the Odyssey DLx Imaging System (LI-COR Biosciences). Immunoblot band intensity quantification was performed using ImageStudio Lite Ver 5.2 software (LI-COR Biosciences).

## Biotinylation assays

Biotinylation of cell surface protein was performed as described (*Crupi et al., 2020*; *Reyes-Alvarez et al., 2022*). Briefly, cell surface proteins were labeled with biotin and either harvested immediately (simple cell surface biotinylation assay) or cells were treated with GDNF for the indicated times to induce RET internalization (biotinylation internalization assay). Following internalization, remaining biotin was removed from the cell surface using MeSNa stripping buffer. Cells were lysed and biotinylated proteins immunoprecipitated with streptavidin-conjugated agarose beads (Thermo Fisher Scientific) overnight at 4°C with agitation. Biotin-labeled proteins were collected by centrifugation at 2000×*g*, washed four times with lysing buffer, and resuspended in Laemmli buffer prior to SDS-PAGE, as previously described (*Crupi et al., 2020*).

## Degradation assays

Serum-starved cells were treated with serum-free media containing GDNF and 100 µg/ml CHX for the indicated times. Proteins were collected for immunoblotting, as above.

## Proliferation assays

Cell proliferation was evaluated through a 3-(4,5-dimethylthiazol-2-yl)-2,5-diphenyltetrazolium bromide (MTT) reduction assay. Briefly, $1.5 \times 10^4$ cells/well were seeded in 96-well plates and allowed to proliferate for indicated times. MTT (Sigma-Aldrich) was added to a final concentration of 0.45 mg/ml for 16 hr. Acidified 20% SDS was added, and cells were incubated at room temperature for 4 hr with shaking. Formazan product was quantified by absorbance at 570 nm and normalized to blank wells lacking cells. Absorbance relative to the 0 hr timepoint for each condition indicated relative cell viability.

## Immunofluorescence and microscopy

ARPE-19 cells were seeded onto poly-L-lysine-coated coverslips for transfection. SH-SY5Y cells were seeded onto bovine collagen 1 (Corning, Oneonta, NY, USA) coated coverslips, or into ibiTreat µ-slide VI 0.4 channel slides (Ibidi, Munich, Germany). Cells were fixed with 4% paraformaldehyde, permeabilized with 0.2% Triton X-100, and blocked in 3% BSA. Cells were incubated with the indicated primary antibodies (1:200–1:400) overnight at 4°C, followed by Alexa Fluor/Cyanine secondary antibodies (1:2000) and indicated stains, Phalloidin (1:40), DAPI/Hoechst 33342 (1:5000–1:10,000). Lipid Raft staining was performed with the Vybrant Alexa Fluor 488 Lipid Raft Labelling Kit according to the manufacturer's instructions (Thermo Fisher Scientific). Coverslips or channel slides were then mounted with MOWIOL-DABCO mounting media (Sigma-Aldrich) and stored at 4°C.

PLA was performed using the Duolink PLA reagents according to the manufacturer's instructions (Sigma-Aldrich). An EVOS M7000 imaging system (Thermo Fisher Scientific) with a 40× air objective was used to acquire immunofluorescent images of the PLA. Particles were analyzed using ImageJ software (US National Institutes of Health, Bethesda, MD, USA, RRID:SCR_003070).

A Leica TCS SP8 confocal microscope system (Leica Microsystems, Wetzlar, HE, Germany) with a 63×/1.40 oil objective lens was used to acquire 0.9 µm optical sections. Images were deconvoluted and ImageJ software (US National Institutes of Health) was used to analyze colocalization and normalize channel intensities to generate the final images.

TIRFM was performed using a Quorum (Guelph, ON, Canada) Discovery microscope, comprised of a Leica DMi8 microscope equipped with a 63×/1.49 NA TIRF objective with a 1.8× camera relay (total magnification ×108). Images were acquired using a Zyla 4.2-megapixel sCMOS camera (Oxford

Instruments) with 488, 561, or 637 nm laser illumination and 527/30, 630/75, or 700/75 emission filters. Fixed-cell TIRFM imaging was performed at room temperature with samples mounted in PBS. Live-cell imaging experiments were performed with cells at constant 37°C during imaging, in phenol red-free, serum-free DMEM media (Gibco, Thermo Fisher Scientific) supplemented with 20 mM HEPES, with imaging at 1 frame-per-second for 5 min.

## Dual iterative SIM
### Imaging
TMEM127-KO or Mock-KO cells stably expressing EGFP-CLC were grown on #1.5 coverslips. Cells were fixed in 4% paraformaldehyde and mounted in Prolong Glass mounting medium (Thermo Fisher, Waltham, MA, USA). Cells were imaged with a Plan-Apochromat 63×/1.4 Oil immersion objective on an ELYRA7 microscope in lattice SIM mode (Carl Zeiss, White Plains, NY, USA). A 488 nm laser was passed through a 23 μm grating and used to excite EGFP. Fluorescence emission was collected on a sCMOS camera (PCO, Kelheim, Germany) through a 495–550 nm bandpass emission filter. 15 illumination phases were collected with an exposure time of 100 ms each. Voxel dimensions were 0.063, 0.063, and 0.110 nm (X, Y, Z), respectively. Z-stacks of approximately 750 nm were acquired for each field of view.

### Processing
3D diSIM processing was performed in ZEN Black 3.0 SR edition (Carl Zeiss White Plains, NY, USA) using the 'Standard Fixed' setting. For initial visual inspections, maximum intensity projections of ~740 nm Z-stacks were performed in ImageJ/Fiji. Z-stacks encompassing entire cells were segmented using the 'Blobs Finder' tool in Vision4D (Aviris; Rostock, Germany) and rendered in 3D. For quantitative analysis of CCP diameter and number, single Z-planes in or near the lower plasma membrane were selected and run through a custom ImageJ/Fiji macro that counted CCPs and determined their 2D area.

## TIRFM experiment analysis
### Fixed-cell analysis of RET colocalization with CLSs
Unbiased detection and analysis of clathrin diffraction-limited structures was performed using CME analysis software in Matlab (Mathworks Corporation, Natick, MA, USA, RRID:SCR_001622; https://github.com/Danuserlab/CMEAnalysis, *DanuserLab, 2024*), as previously described (*Aguet et al., 2013*; *Cabral-Dias et al., 2022*; *Delos Santos et al., 2017*). Briefly, diffraction-limited clathrin structures were detected using a Gaussian-based model to approximate the point-spread function of CLSs ('primary' channel). RET proteins (TIRFM images) or clathrin (epifluorescence images) were detected in a 'secondary' channel and the intensity within the CLSs, beyond local background amounts, signified the level of colocalization of the proteins.

### Live-cell analysis of CCP dynamics
Automated detection, tracking, and analysis of CCPs following time-lapse TIRF microscopy of SH-SY5Y cells stably expressing EGFP-CLC was as previously described (*Aguet et al., 2013*; *Kadlecova et al., 2017*; *Mettlen and Danuser, 2014*). Diffraction-limited clathrin structures were detected using a Gaussian-based model method to approximate the point-spread function (*Aguet et al., 2013*), and trajectories were determined from clathrin structure detections using u-track (*Jaqaman et al., 2008*). sCLSs were distinguished from bona fide CCPs based on unbiased analysis of clathrin intensities in the early CCP stages (*Aguet et al., 2013*; *Kadlecova et al., 2017*). Both sCLSs and CCPs represent nucleation events, but only bona fide CCPs undergo stabilization, maturation, and in some cases scission to produce vesicles (*Aguet et al., 2013*; *Kadlecova et al., 2017*). We report the sCLS nucleation, CCP initiation, and the fraction of CCPs <15 s. Because CCPs are diffraction-limited objects, the amplitude of the Gaussian model of the fluorescence intensity of EGFP-CLC informs about CCP size (*Figure 5A–E*).

## Single-particle tracking
Single-particle tracking experiments were performed by labeling EGFR with a Cy3B-conjugated Fab fragment targeting the EGFR ectodomain, generated from mAb108 (*Sugiyama et al., 2023*). Previous

characterization of this labeling strategy indicated that antibody labeling did not impact EGFR ligand binding or signaling (*Sugiyama et al., 2023*). Cells were labeled with 50 ng/ml Fab-Cy3B (to label and track total EGFR) for 10 min followed by washing to remove unbound antibodies, in media lacking EGF. Time-lapse TIRF imaging was performed using a Quorum (Guelph, ON, Canada) Diskovery system, comprised of a Leica DMi8 microscope with a 63×/1.49 NA TIRF objective with a 1.8× camera relay (total magnification ×108), using 561 nm laser illumination and a 620/60 nm emission filter. Images were acquired using an iXon Ultra 897 EM-CCD (Spectral Applied Research, Andor, Toronto, ON, Canada). All live-cell imaging was performed using cells incubated in serum-free DMEM/F-12 or DMEM without phenol red or P/S, with a frame rate of 20 Hz, for total length of time-lapse 250 frames. Single particles labeled with Cy3B were detected and tracked (*Jaqaman et al., 2008*), and the motion types and diffusion coefficients were determined using moment-scaling spectrum analysis (*Freeman et al., 2018*; *Jaqaman et al., 2011*).

## Statistical analyses

A minimum of three independent replicates were performed for each experiment, except where indicated. Consistency between experimental replicates was assessed through two-tailed unpaired t-tests, and one- or two-way ANOVA with indicated multiple comparison tests using GraphPad Prism version 9 (GraphPad Prism Software, San Diego, CA, USA). Specific significance tests were selected using GraphPad Prism following residual testing for Gaussian distribution (normality tests of Anderson-Darling, D'Agostino, Shapiro-Wilk, Kolmogorov-Smirnov) and testing whether residuals are clustered or heteroscedastic (Brown-Forsyth and Bartlett's tests). Bar and line graphs, used for immunoblots and time course experiments, were presented with means and standard deviation (± SD), while box plots with whiskers from minimum to maximum values were used for individual cell imaging. Graphs were produced using GraphPad Prism.

## Acknowledgements

The authors would like to thank Juliana Shizas, Montdher Hussain, and Kasra Ranjbar for technical assistance. This work was supported by operating grants from the Cancer Research Society of Canada (CRS-19439), Canadian Institutes for Health Research (MOP-142303, PJT-178274, LMM) and (PJT-156355, CNA), and from NIH (GM114102 and CA264248), Neuroendocrine Tumor Research Foundation, VHL Alliance, and the UT System Star Awards (PLMD). PLMD holds the Robert Tucker Hayes Distinguished Chair in Oncology. LMM is the Bracken Chair in Genetics and Molecular Medicine at Queen's University.

## Additional information

### Funding

| Funder | Grant reference number | Author |
| --- | --- | --- |
| Cancer Research Society | CRS-19439 | Lois M Mulligan |
| Canadian Institutes of Health Research | MOP-142303 | Lois M Mulligan |
| Canadian Institutes of Health Research | PJT-156355 | Costin N Antonescu |
| National Institutes of Health | GM114102 | Patricia LM Dahia |
| Canadian Institutes of Health Research | PJT-178274 | Lois M Mulligan |
| National Institutes of Health | CA264248 | Patricia LM Dahia |

The funders had no role in study design, data collection and interpretation, or the decision to submit the work for publication.

## Author contributions

Timothy J Walker, Conceptualization, Data curation, Software, Formal analysis, Validation, Investigation, Visualization, Methodology, Writing – original draft, Project administration, Writing – review and editing; Eduardo Reyes-Alvarez, Conceptualization, Software, Formal analysis, Investigation, Visualization, Methodology, Writing – review and editing; Brandy D Hyndman, Resources, Supervision, Validation, Investigation, Methodology, Writing – review and editing; Michael G Sugiyama, Data curation, Software, Formal analysis, Investigation, Visualization, Methodology, Writing – review and editing; Larissa CB Oliveira, Data curation, Formal analysis, Validation, Investigation; Aisha N Rekab, Conceptualization, Investigation; Mathieu JF Crupi, Conceptualization, Methodology, Writing – review and editing; Rebecca Cabral-Dias, Resources, Supervision; Qianjin Guo, Patricia LM Dahia, Resources; Douglas S Richardson, Conceptualization, Resources, Data curation, Formal analysis, Investigation, Visualization, Methodology, Writing – review and editing; Costin N Antonescu, Conceptualization, Resources, Software, Supervision, Funding acquisition, Methodology, Writing – review and editing; Lois M Mulligan, Conceptualization, Resources, Supervision, Funding acquisition, Writing – original draft, Project administration, Writing – review and editing

## Author ORCIDs

Timothy J Walker ⓘ https://orcid.org/0000-0002-6146-3485
Brandy D Hyndman ⓘ https://orcid.org/0000-0003-4096-0922
Larissa CB Oliveira ⓘ https://orcid.org/0000-0001-6798-0662
Patricia LM Dahia ⓘ https://orcid.org/0000-0002-7757-370X
Douglas S Richardson ⓘ http://orcid.org/0000-0002-3189-2190
Costin N Antonescu ⓘ https://orcid.org/0000-0001-9192-6340
Lois M Mulligan ⓘ https://orcid.org/0000-0002-9308-8245

Reviewer #4 (Public review): https://doi.org/10.7554/eLife.89100.3.sa1
Reviewer #5 (Public review): https://doi.org/10.7554/eLife.89100.3.sa2
Author response https://doi.org/10.7554/eLife.89100.3.sa3

---

# Additional files

## Supplementary files

• MDAR checklist

## Data availability

All data generated during this study are included in the manuscript and supporting files. Source data files have been provided.

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
