## [Editor Report · eLife assessment]

This **valuable** paper provides **convincing** evidence that loss of the tumor suppressor TMEM127 causes disorganization of plasma membrane lipid domains, alters clathrin assembly, and inhibits endocytosis of a variety of cell surface receptors, leading to increased cell surface levels of signaling proteins including RET and other transmembrane receptor tyrosine kinases. The results are significant for understanding how RET127 loss contributes to pheochromocytoma, although the evidence is indirect owing to the lack of human pheochromocytoma cell lines. The results will be of interest for researchers studying pheochromocytoma and endocytosis mechanisms.

---

## [Referee Report · Reviewer #4 (Public review)]

Summary:

Walker et al. investigated the function of TMEM127 on RET regulation and function that could contribute to the development of pheochromocytoma (PCC). The authors showed that deletion of TMEM127 causes RET protein accumulation on the cell surface and, thereby, increased its constitutive ligand-independent activity and downstream signaling. They also unveiled the mechanism of how TMEM127 regulates cell membrane dynamics, particularly focusing on clathrin distribution and its effects on cargo internalization.

Strengths:

They showed that the deletion of TMEM127 affected multiple classes of transmembrane proteins, including RTKs (RET, EGFR), cell adhesion molecules (N-Cadherin, Integrin Beta-3), and carrier proteins (Transferrin Receptor-1), suggesting a global effect on cell surface proteins. This, at least in part, may explain how TMEM127 mutations act as drivers in PCC as well as in other cancers, such as renal cell carcinoma, where RET is not highly expressed. Overall, these findings provide new insights into the understanding of pheochromocytoma pathogenesis and potentially other cancers.

Weaknesses:

The major weakness of this study is the lack of human PCC cell lines for evaluating the function of TMEM127. Currently, the cell line models for pheochromocytoma are unavailable, and manipulation of patient-derived organoids is challenging. To complement this weakness, they provided immunohistochemical data showing that RET protein is highly expressed in TMEM127-mutant PCC.

Furthermore, some of the authors in this manuscript recently published a paper titled 'TMEM127 suppresses tumor development by promoting RET ubiquitination, positioning, and degradation' (Guo et al. Cell Reports 42, 113070, 2023, which is also cited in the current manuscript). In this manuscript, they showed that TMEM127 binds to RET and recruits the NEDD4 E3 ubiquitin ligase for RET ubiquitination and degradation via TMEM127. In general, the ubiquitination of proteins is highly specific to each molecule. In the current version of the manuscript, there is no description of the relevance between these two potentially different mechanisms (clathrin-mediated or ubiquitin-mediated) of accumulating RET and/or other proteins mentioned in two separate papers. I believe the authors should at least discuss this.

---

## [Referee Report · Reviewer #5 (Public review)]

Summary:

The manuscript by Walker et al., nicely demonstrated a role of TMEM127 in regulating membrane dynamics of RET, a receptor tyrosine kinase and oncogene for multiple cancers, particularly in pheochromocytoma (PCC). They provided compelling cellular and biochemical evidence on how TMEM127 deficiency reduces RET internalization and degradation in specific cancer cell lines, thus stabilizing cell surface RET and promoting its signaling related to cell proliferation. Moreover, TMEM127 may have a broad function beyond RET, and may affect other surface protein activity such as EGFR etc. These findings provided novel mechanisms and key insights to the field of cancer biology.

Strengths:

Very interesting findings that nicely explained the mechanistic link between TMEM127 and tumorigenesis by RET regulation...the biochemical analysis was quite thorough and impressive.... the general messages delivered by this study are considered as important to the field of TMEM127 biology and tumorigenesis.

Weaknesses:

As acknowledged by the authors, the role of TMEM127 can be conditional and beyond RET modulation, the authors may need to include more discussion that why the loss of TMEM127 is more linked to the development of PCC compared to other cancer types, and why TMEM127 can have such a broad effects in those membrane molecules...in addition, TMEM127 deficiency has been recently linked to enhanced MHC-I-mediated tumor immunity and tumor eradication, in some cancer types...it is then worthwhile to discuss the dual effects of TMEM127 in tumor control in the context of immunity...

Moreover, the authors may need to tune down their "ligand independent" claim on the loss of TMEM127 in driving RET signaling, which should be more related to the level of RET expression/strength of clustering...

---

## [Author Response]

The following is the authors’ response to the original reviews.

This study highlights new insights into the mechanism of pheochromocytoma pathogenesis that remains poorly understood. In the context of hereditary syndromes, such as multiple endocrine neoplasia 2 (MEN-2), where RET mutation is the major driver of thyroid, parathyroid, and adrenal pathologies, including pheochromocytoma, this mechanistic dissection of RET and TMEM127 is fundamentally sound. While the significance was deemed important, the strength of the evidence was found to be solid,

Recognizing the limitations of models available for study of neuroendocrine cancers, and specifically for pheochromocytomas, we have revised and clarified the text of the current manuscript version and provide specific responses to the additional comments provided below, highlighting changes and new data.

**Reviewer #1 (Recommendations For The Authors):**
A current lack of pheochromocytoma cell lines and the use of generated cell lines for mechanistic studies presents a significant challenge that may undermine the inferred value of these findings in mock in vitro systems and question reproducibility in pheochromocytoma. Consideration for 3-dimensional patient-derived pheochromocytoma organoid in vitro and patient-derived organoid xenograft in vivo models will enable confirmation or refute novel findings described by the authors.

We agree completely with Reviewer 1 that ideally, we should replicate these findings with PCC-derived cells in vitro and in organoids. Despite many attempts, PCC cell lines have proved a major challenge for the field of neuroendocrine cancers. Cell line models are not available and PDOs have proven poorly growing and resistant to manipulations, such as CRISPR KOs or siRNA KD. In studies completed since the submission and review of the present manuscript, and subsequently published elsewhere, we have shown that RET protein is highly expressed in TMEM127-mutant PCC by immunohistochemistry. We also showed that the TMEM127-KO SH-SY5Y cell model does grow more robustly than Mock-KO cells in nude mice and that RET inhibition (Selpercatinib) does lead to tumor regression (Guo et al., 2023), suggesting that our findings may be reproducible in vivo. These findings, and potential caveats of the cell models used have been further discussed in the text.

**Reviewer #2 (Recommendations For The Authors):**
Most notably, all experiments are conducted in an isogenic single-cell line. This exposes the whole story to be potentially confounded by unknown variables.In addition, studies would benefit from the adding back of TMEM127, or other methods to modulate endosome and plasma membrane dynamics to mechanistically secure the cause of the findings.

As suggested by Reviewer 2, we have generated a TMEM127 KO in HEK293, an unrelated cell line which expressed low levels of TMEM127 but does not express RET. Consistent with our findings in SH-SY5Y, we saw increased membrane accumulation of endogenous membrane proteins N-cadherin and transferrin receptor-1 in these cells in the absence of TMEM127. Additionally, re-expression of a wildtype TMEM127 (FLAG-TMEM127) in these cells led to dramatic decreases in membrane localization of these proteins (Supplemental Figure 1D). These data suggest that membrane accumulation is indeed TMEM127 dependent, and that these processes are not directly dependent on RET expression.

References

Guo, Q., Z.M. Cheng, H. Gonzalez-Cantu, M. Rotondi, G. Huelgas-Morales, P. Ethiraj, Z. Qiu, J. Lefkowitz, W. Song, B.N. Landry, H. Lopez, C.M. Estrada-Zuniga, S. Goyal, M.A. Khan, T.J. Walker, E. Wang, F. Li, Y. Ding, L.M. Mulligan, R.C.T. Aguiar, and P.L.M. Dahia. 2023. TMEM127 suppresses tumor development by promoting RET ubiquitination, positioning, and degradation. Cell Rep. 42:113070.